# Atmospheric methane since the LGM was driven by wetland sources

Thomas Kleinen[1], Sergey Gromov[2], Benedikt Steil[2], and Victor Brovkin[1]

[1]Max Planck Institute for Meteorology, Bundesstr. 53, 20146 Hamburg, Germany
[2]Max Planck Institute for Chemistry, Hahn-Meitner-Weg 1, 55128 Mainz, Germany

**Correspondence:** Thomas Kleinen (thomas.kleinen@mpimet.mpg.de)

**Abstract.** Atmospheric methane ($CH_4$) has changed considerably in the time between the last glacial maximum (LGM) and the preindustrial period (PI). We investigate these changes in transient experiments with an Earth System Model capable of simulating the global methane cycle interactively, focusing on the rapid changes during the deglaciation, especially pronounced in the Bølling Allerød (BA) and Younger Dryas (YD) periods. We consider all relevant natural sources and sinks of methane and examine the drivers of changes in methane emissions as well as in the atmospheric lifetime of methane. We find that the evolution of atmospheric methane is largely driven by emissions from tropical wetlands, while variations in $CH_4$ atmospheric lifetime are not negligible but small. Our model reproduces most changes in atmospheric methane very well, with the exception of the mid-Holocene decrease in methane, though the timing of ice sheet meltwater fluxes needs to be adjusted slightly in order to exactly reproduce the variations of the BA and YD.

## 1 Introduction

Between the last glacial maximum (LGM) and the present, the atmospheric concentration of methane ($CH_4$) changed dramatically. At the LGM, atmospheric $CH_4$ was at ~380 $ppb$ (Köhler et al., 2017), while it was at 695 $ppb$ at 10 ka BP, thus nearly doubling in concentration during those 11000 yrs (Fig. 2). Furthermore, the atmospheric concentration changed very rapidly at three points in time: during the transition from the Oldest Dryas into the Bølling Allerød (BA), during the transition from the BA into the Younger Dryas (YD), and from the YD to the Preboreal (PB) / early Holocene. During each of these transitions atmospheric $CH_4$ changed by about 150 $ppb$ over a few centuries. Finally, atmospheric methane has more than doubled again between the preindustrial state (1850 CE) and the present (2022 CE). In a previous publication, we investigated the methane emissions during a number of time slices from 20 ka BP to the present (Kleinen et al., 2020). Time slice investigations, however, are very limited in their explanatory power. They rest on the assumption that the system under investigation is in some kind of an equilibrium state, and they give no information at all on how the system might translate between these states. In order to gain insight into highly dynamic changes, transient experiments considering the full dynamics of the system therefore are required. Here, we present the very first transient deglaciation experiments with a state-of-the-art Earth System Model (ESM) including a complete methane cycle.

Attempts at modelling methane between the LGM and PI go back several decades and mostly fall into one of two categories. Many of these studies were performed with strongly simplified models, typically box models, that only consider very broad spatial aggregations and an extremely reduced process description, for example Chappellaz et al. (1993); Thompson et al.

(1993); Fischer et al. (2008). The alternative to very simplified models were studies using models with more detailed process descriptions of at least part of the methane cycle, ranging from the wetland emissions of methane to a more or less complete description of the methane cycle, for example Valdes et al. (2005); Weber et al. (2010); Singarayer et al. (2011); Zürcher et al.

(2013); Hopcroft et al. (2017); Kleinen et al. (2020). These studies were, however, limited to a few time slices, assuming an equilibrium in climate for these time slices and not covering the trajectories connecting these points in time. Here, we aim to go beyond this state of the art, performing transient experiments with a full ESM specifically adapted to cover the glacial period and the transitions between glacial and interglacial.

The requirements to comprehensively investigate the changes in methane from the LGM to PI are demanding. Ice sheet extent

changes by several million $km^2$, sea level changes by about $130\,m$ (Lambeck et al., 2014), and atmospheric $CO_2$ increases by roughly 50% from $180\,ppm$ to $280\,ppm$ (Petit et al., 1999). These processes, while affecting methane only indirectly, need to be represented in the ESM in order to reproduce the climatic changes driving the changes in the methane cycle. For the methane cycle itself, the global methane budget (Saunois et al., 2016, 2020) considers emissions of fossil fuels as well as agriculture and waste as anthropogenic emissions. These do not need to be considered here, as our experiments mainly consider the time

before anthropogenic emissions become relevant. However, emissions from wetlands and wildfires, as well as several other sources lumped under the 'other natural emissions', namely termites and wild animals, need to be considered, together with the sinks of methane in the atmosphere and soils.

We investigated methane emissions for time slices spaced every 5000 years, starting at the LGM and ending at the present in Kleinen et al. (2020). We also showed the evolution of atmospheric $CH_4$ for the next millennium under a number of scenarios

in Kleinen et al. (2021). In the present publication we bring these two together and investigate the transient evolution of the methane cycle from the LGM to PI, mainly focusing on the period with the largest changes in climate and methane from 18 ka BP to 10 ka BP, which includes the fast and massive changes during the Bølling-Allerød and Younger Dryas.

## 2   Modelling the methane cycle in MPI-ESM 1.2

### 2.1   MPI-ESM 1.2 in transient deglaciation experiments

We use MPI-ESM, the Max Planck Institute for Meteorology Earth System Model (Mauritsen et al., 2019; Mikolajewicz et al., 2018), consisting of the atmospheric general circulation model ECHAM6, the ocean general circulation model MPI-OM, and the land surface model JSBACH in coarse resolution (T31GR30 $\approx$ 3.75° x 3.75°) to investigate the evolution of atmospheric methane during the deglaciation. The basis for our modelling setup is model version 1.2, the same model version as used in the Coupled Model Intercomparison Project Phase 6 (CMIP6, Eyring et al. (2016)).

MPI-ESM, similar to many other ESMs, considers the land-sea and glacier masks, as well as the river routing, as fixed. For determining the transient evolution of climate, these are updated automatically in our model version, as described in Meccia and Mikolajewicz (2018) and Riddick et al. (2018). Briefly, the model evaluates the changes in ice sheet and sea level after every decade. From these, combined with the RTopo-2 topography (Schaffer et al., 2016), it determines a new land-sea mask, bathymetry and orography, as well as a new river routing setup to be used for the next decade in the model experiment.

## 2.2 Modelling the methane cycle

Based on the recent Global Carbon Project $CH_4$ budget (Saunois et al., 2016, 2020), we consider methane emissions from wetlands, termites, wildfires and herbivores as relevant during the deglaciation, as well as emissions from geological sources. We assume the latter to be independent of climate and therefore prescribe them, while we aim to model all of the former interactively. We neglect anthropogenic methane emissions in the experiments described here, as they are highly uncertain but likely very small for the time period considered here. In terms of methane sinks, the atmospheric sink of methane is most important, while soils are a secondary sink. We include both of them in our model.

### 2.2.1 Methane emissions from wetlands, termites, wildfires and geological sources

The basic methane emission model for emissions from wetlands, termites, and wildfires is described in Kleinen et al. (2020), we thus cover it only briefly here. We use a TOPMODEL (Beven and Kirkby, 1979) approach to determine inundated areas, in which we determine the wetland methane emissions based on the model by Riley et al. (2011). Methane emissions from fires are determined using the SPITFIRE fire model (Lasslop et al., 2014), employing emission factors by Kaiser et al. (2012). Termite methane emissions are estimated using the approach from Kirschke et al. (2013) and Saunois et al. (2016).

In the TOPMODEL approach (Beven and Kirkby, 1979), we combine the soil water content determined in the MPI-ESM land surface model JSBACH with sub-gridscale topographic information, the Compound Topographic Index (CTI), in order to determine the variation of the water table in each model grid cell. For our experiments, we use the CTI index product by Marthews et al. (2015) for the present-day land areas, combined with CTI index values we derived from the the ETOPO1 dataset (Amante and Eakins, 2009) for those areas that are below sea level at present, using the TOPMODEL R library (Buytaert, 2011). We then use the variation in water table to determine the inundation fraction, the fraction of each grid cell where the water table is at or above the surface. In Kleinen et al. (2020) we have evaluated the model for present-day climatic conditions against remote-sensing data of inundation (Prigent et al., 2012). Total extent and seasonality are similar for the NH extratropics, while the model slightly overestimates the extent for tropical inundation. We thus assess the agreement between model and data as reasonable, considering the limitations of both model and remote-sensing data.

In JSBACH the YASSO model (Goll et al., 2015) is used to determine the decomposition of soil carbon. In the inundated areas we assume anaerobic decomposition, with decomposition rate constants reduced to 35% of the YASSO values used for aerobic decomposition, as proposed by Wania et al. (2010). The anaerobic decomposition of carbon produces both $CO_2$ and $CH_4$, with a temperature-dependent partitioning into the two decomposition products as described by Riley et al. (2011). The transport of $O_2$, $CO_2$ and $CH_4$ is determined in a methane emission model based on Riley et al. (2011), which explicitly simulates the methane transport via the pathways diffusion, ebullition, and plant aerenchyma. The oxidation of $CH_4$, if sufficient oxygen is present, is considered as well, following Michaelis-Menten kinetics. Furthermore the transport model also determines the soil sink of methane, as $CH_4$ diffuses into the soil in areas where little methane is produced, i.e. in dry areas, where it is oxidized subsequently.

Lake methane emissions are not modelled explicitly, we rather assume that their emissions are implicitly contained in the wetland flux. We make this simplification for pragmatic reasons, as we have not found an appropriate model for lake areas under changing climatic conditions in the literature. We assume, however, that the error introduced by this simplification is relatively minor on the scales the model was designed for (~350+ km spatial resolution, decadal to centennial temporal scale). We base this assumption on two factors: We used surface water extent data to calibrate the wetland model, and this data also contains inland water bodies. Furthermore, we assume that the changes in methane fluxes from inland waters on these scales will be driven by the same factors that drive the changes in wetland emissions, i.e. soil carbon content, temperature, and precipitation. On shorter temporal (monthly to annual) and spatial (10s of kilometers) scales, the errors introduced through this simplification may be significant, though.

Methane emissions from wildfires and biomass burning (with the sum subsequently called the 'fire' emissions) are determined from the SPITFIRE fire model (Lasslop et al., 2014), using emission factors from Kaiser et al. (2012). The SPITFIRE fire model determines the spread of fires using the fire ignition probability, a function of lightning frequency and population density (assuming negligible human impact on fires before 12 ka BP due to small population size and using Klein Goldewijk et al. (2017) afterwards), and flammability (higher under dryer/warmer conditions), as well as the amount of biomass available for burning. The methane emissions are then determined from the burned biomass using emission factors. Termite methane emissions are estimated using the approach by Kirschke et al. (2013) and Saunois et al. (2016), which determines termite mass from gross primary productivity in tropical areas and assumes a constant emission factor to determine the final methane emissions. Finally, methane emissions from geological sources are prescribed using a spatial distribution from Etiope (2015), but scaled down to give total geological methane emissions of $5\,TgCH_4\,yr^{-1}$, as Petrenko et al. (2017) and Hmiel et al. (2020) show from ice-core data that geological emissions larger than this value are not possible for either the Younger Dryas or the preindustrial period.

In Kleinen et al. (2020) we evaluated the modelled methane emissions for present-day (PD) climate. As flux measurements on appropriate scales are not available, we compared aggregate fluxes against global assessments (Saunois et al., 2016). We found that the model simulates wetland methane emissions of $222\,TgCH_4\,yr^{-1}$ (decadal mean over 2000-2009), reduced to $190\,TgCH_4\,yr^{-1}$ in this study as detailed below, fire emissions of $17.6\,TgCH_4\,yr^{-1}$, termite emissions of $11.7\,TgCH_4\,yr^{-1}$, and a soil uptake of $17.5\,TgCH_4\,yr^{-1}$. These values fall well within the ranges reported by Saunois et al. (2016), who report $153-227\,TgCH_4\,yr^{-1}$ for natural wetlands, $15-20\,TgCH_4\,yr^{-1}$ for biomass burning, $1-5\,TgCH_4\,yr^{-1}$ for wildfires, $3-15\,TgCH_4\,yr^{-1}$ for termites, and $9-47\,TgCH_4\,yr^{-1}$ for the soil uptake. Spatial patterns of PD emissions are also similar to those shown by Saunois et al. (2016). Furthermore, wetland methane emission estimates from atmospheric inversions (Bousquet et al., 2011) show that the majority (62-77%) of the present-day emissions come from regions between 30°S and 30°N, while a much smaller part (20-33%) is emitted north of 30°N. Of the modelled total wetland $CH_4$ emissions for PD conditions, 70% are from low latitude regions, while 29% are from regions north of 30°N. The latitudinal distribution of modelled PD wetland methane emissions therefore is well within the range obtained from atmospheric inversions.

In order to accommodate the additional methane flux from the consideration of herbivorous mammals in the present publication, we re-calibrated the wetland emission model in comparison to Kleinen et al. (2020), reducing wetland emissions to

$190\,Tg\,CH_4\,yr^{-1}$ for the 2000-2009 CE decadal mean, thus keeping total emissions for the present-day roughly constant, despite the additional consideration of herbivore emissions.

### 2.2.2 Methane emissions from herbivores

At present, methane emissions from herbivores, especially cows, make up a significant part of the methane emissions (Saunois et al., 2020). This fact is, however, the result of human action, as it has to be assumed that the number of ruminants was significantly smaller before humans began herding cows. Previous approaches to estimate the methane emissions from herbivores (Crutzen et al., 1986; Chappellaz et al., 1993; Hristov, 2012; Smith et al., 2016) generally start from the level of individual animals, relating methane emissions to species and body mass. They then rely on estimates of population numbers for specific species to determine total emissions from herbivores. In order to apply such an approach in the context of an Earth System model and past climate states, one would need to somehow relate population densities of certain species via vegetation productivity to climate – which we found impossible to do due to a lack of data, especially for ecosystems untouched by humans, as none of these exist any more. Furthermore, a recent review of methane production by mammal herbivores (Clauss et al., 2020) found that $CH_4$ yields ($CH_4$ production per dry matter (DM) intake) do not vary significantly with body mass and between ruminants and non-ruminants, thus negating two of the foundations of the previous approaches to estimating herbivore methane emissions. Instead, they found that absolute $CH_4$ emissions scaled linearly with DM intake, which allows a simplified treatment of herbivore methane emissions in our model.

As part of the carbon cycle representation, the JSBACH model determines a carbon flux $F_{herbivory}$ from herbivory (Schneck et al., 2013; Reick et al., 2021). Here, $F_{herbivory} = r_{herbivory} \times C_G$ with $r_{herbivory}$ a constant dependent on the plant functional type and $C_G$ the carbon content in the 'green' carbon pool, i.e. the carbon pool representing the living parts of plants (leaves, fine roots, vascular tissues). We further assume that a fraction $f_{mammal}$ of the herbivory flux is consumed by mammals, with $f_{mammal} = 0.016$ in forest and $f_{mammal} = 0.32$ in grasslands. These latter values were chosen ad hoc but seem plausible. Finally, we use a $CH_4$ yield $\gamma$ of $\gamma = 14.9\,g(CH_4)\,kg(DM)^{-1}$, obtained as a mean over the values for all species listed in Clauss et al. (2020). The final $CH_4$ flux from herbivory $H_{CH4}$ thus is:

$$H_{CH4} = f_{mammal} \times \gamma \times F_{herbivory} \tag{1}$$

### 2.3 Atmospheric methane sink

As described in Kleinen et al. (2021), the spatiotemporal evolution of the methane abundance in the atmospheric model ECHAM6 is simulated using a methane tracer which undergoes transport and chemical removal, while emissions are calculated using the land surface model JSBACH. The atmospheric sink of methane is calculated using a zonally averaged reactivity field obtained from the comprehensive ECHAM/MESSy Atmospheric Chemistry Model (EMAC) (Joeckel et al., 2010). This reactivity field contains, in aggregated form, all atmospheric sink terms considered in the EMAC model, i.e., reaction with OH, tropospheric Cl, stratospheric reactions etc.

Following Gromov et al. (2018) and Kleinen et al. (2021), the following updated parameterisation is used to account for variations in atmospheric oxidative capacity and therefore tropospheric $CH_4$ reactivity $r_{CH_4}$:

$$160 \quad r_{CH_4} = \alpha \times (LN + k_N RN)^p \times (M + k_C RC + k_A A)^q \, [yr^{-1}] \tag{2}$$

with $LN$ being the global lightning nitrogen oxides ($NO_X$) emission, simulated interactively according to Price and Rind (1992, 1993), $M$ the $CH_4$ atmospheric burden, $RN$ and $RC$ the terrestrial (surface) emissions of reactive nitrogen ($NO_X$) and carbon compounds given in $TgN$ and $TgC$ per $yr$ for the emission fluxes, respectively, and $A$ the mean tropospheric reactive aerosol surface area (about $100 Mm^2$ in both present-day and past climates). Fit parameters ($\alpha = 7.45 \, TgN^{-p} TgC^{-q} yr^{-1}$, $p = 0.36$, $q = -0.60$, $k_N = 0.59$, $k_C = 4.25$, $k_A = 4.21 \, TgC/Mm^2$) are obtained from an ensemble of EMAC simulations covering a broad range of $RN$, $RC$, $LN$ and $M$ values in the LGM (21 ka BP), Mid-Holocene (6 ka BP), PI and present-day conditions. The fitted $r_{CH_4}$ value is accurate to 7% at 95% confidence intervals (CI). In the MPI-ESM experiments, the natural emission components of $RN$ and $RC$ are obtained from the MEGAN model (Guenther et al., 2012) for the biogenic sources and from the SPITFIRE model (Lasslop et al., 2014), with emission factors from Kaiser et al. (2012), for fire emissions. We use the terrestrial $NO_X$ emissions for the $RN$ term, and for the $RC$ term we use biogenic CO and isoprene ($C_5H_8$) fluxes, scaled by a factor of 1.4 to account for secondary biogenic co-emitted compounds, as proxies for the total $RC$ emitted. These scaling factors were derived from the simulated present-day total RC emissions. In experiments covering the historical and future periods, anthropogenic emissions of $RC$ and $RN$ would be considered as well (see Kleinen et al. (2021)), but are neglected for the experiments described below.

## 2.4 Model forcing and experiments

We forced the model with prescribed orbital parameters from Berger (1978) and greenhouses gases from Köhler et al. (2017). Orbital parameters and greenhouse gas concentrations are supplied to the model as decadal means and are updated every 10 model years. Atmospheric aerosols were prescribed to constant 1850 conditions (Kinne et al., 2013), and we considered no anthropogenic land use. Ice sheet extent was prescribed from the GLAC-1D ice sheet reconstruction (Tarasov et al., 2012; Briggs et al., 2014; Ivanovic et al., 2016). Ice sheet extent, as well as bathymetry and topography (Meccia and Mikolajewicz, 2018) and river routing (Riddick et al., 2018) were continuously updated every 10 model years.

We initialised the model from a spinup experiment at constant 26 ka BP boundary conditions, running for several millennia. From this model state, we performed a transient model experiment until preindustrial times, with continuously updated ice sheet extent, bathymetry, topography, and river routing, similar to the experiments described by Kapsch et al. (2022). In this experiment, called *base* in the following, the prescribed ice sheet forcing from the GLAC-1D reconstruction leads to a collapse of the Atlantic Meridional Overturning Circulation (AMOC) early in the Bølling-Allerød (Kapsch et al., 2022), due to excess meltwater entering the North Atlantic, and the Younger Dryas does not occur in this experiment. Thus we performed a second experiment, called *MWM* (meltwater manipulation), where we manipulate the meltwater fluxes from the Laurentide ice sheet: Starting in 15.2 ka BP, we prevent meltwater from the Laurentide ice sheet from reaching the ocean, instead storing it. We then release this accumulated meltwater over a period of 1200 years, starting in 12800 BP, adding it to the Mackenzie River

watershed, thus mimicking the storage of glacial meltwater in proglacial lakes like Lake Agassiz and its subsequent release (Murton et al., 2010).

## 3 Results

### 3.1 Transient climate and land carbon changes

At the last glacial maximum, around 20 ka BP, the global mean temperature is $282K$ in our model, $4.7K$ lower than at PI (Fig. 1a). As the ice sheets melt and $CO_2$ increases, the temperature begins to rise significantly after 18 ka BP, with the rate of temperature increase declining after 9 ka BP. Dallmeyer et al. (2022) evaluated model temperatures in the experiment against reconstructions and found a reasonably close match, as illustrated by the comparison against the reconstructions by Shakun et al. (2012) and Osman et al. (2021) in Fig. 1a. While Shakun et al. (2012) reconstructed a smaller temperature change between LGM and Holocene than we see in our experiments, Osman et al. (2021) reconstructed a larger temperature change, with our experiment situated between the two. The Atlantic Meridional Overturning Circulation (AMOC) (Fig. 1b) is relatively stable at about $20\,Sv$ between the LGM and 14.8 ka BP. Here the *base* and *MWM* experiments diverge: in the *base* experiment meltwater pulse (MWP) 1a, a sea level rise event recorded in Barbados corals (Fairbanks, 1989), leads to a strong freshening of the North Atlantic and a near complete AMOC shutdown (remaining overturning of $1\,Sv$) at 14.38 ka BP, which in turn leads to a drop in global mean temperature by nearly $1K$. Temperature and AMOC recover quickly, however. The AMOC continues to be highly variable until 9 ka BP, with a further near collapse from MWP 1b at 11.4 ka BP. Global mean temperature, however, is affected significantly less by the AMOC reduction at MWP 1b than at MWP 1a. After 9 ka BP, the AMOC is relatively stable, though at a slightly smaller overturning of $18.7\,Sv$ (at PI). Similar to global mean temperature, the total land carbon stock (Fig. 1c) stays nearly constant at $970\,PgC$, $718\,PgC$ less than at PI, from LGM to 18 ka BP. Subsequently total land C increases until 10.95 ka BP. Afterwards only minor changes in total land carbon stock occur until the PI state with a total land C stock of $1688\,PgC$. The AMOC collapse at 14.38 ka BP caused by MWP 1a leads to a short decrease in C stock by about 6.8%, but the total duration of this excursion is about 800 years, and carbon continues to rise after it.

In the *MWM* experiment, the initial collapse of the AMOC at 14.38 ka BP is prevented by the manipulation of the meltwater fluxes, and temperature and carbon continue to rise during this period. As we release the accumulated Laurentide ice sheet meltwater at 12.8 ka BP, the AMOC collapses, with the consequence of a drop in global mean temperature by $1.25K$ and a decrease in terrestrial C stock by $100\,PgC$ or 6.9%. After we end the meltwater manipulation in 11.6 ka BP, temperature, AMOC and land carbon stocks recover quickly.

### 3.2 Transient changes in methane

In comparison to a stack of ice-core derived $CH_4$ concentration in Antarctica (Köhler et al., 2017), the atmospheric concentration of methane in our experiments is quite reasonable overall (Fig. 2a), although there are significant discrepancies during some periods. In the ice-core, methane is nearly constant at ~$370\,ppb$ from LGM to 18 ka BP, when it starts increasing slowly.

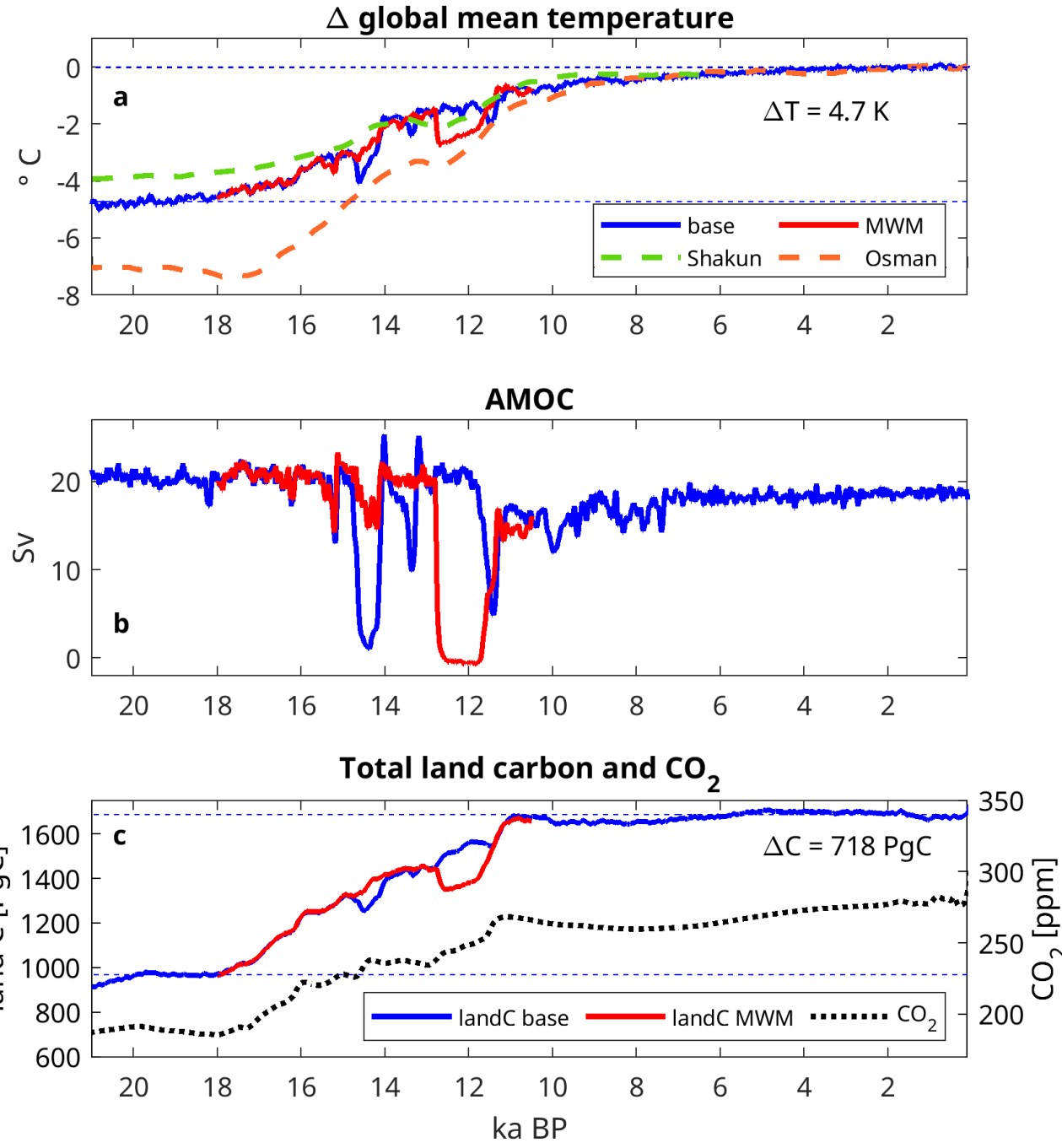

**Figure 1.** Overview over climatic and land carbon changes from LGM to PI: Global mean temperature change (a), North Atlantic meridional overturning circulation (b), total land carbon storage and atmospheric $CO_2$ (c). *Base* experiment in blue, *MWM* experiment in red. (a) also contains temperature change reconstructed by Shakun et al. (2012) and Osman et al. (2021), while reconstructed atmospheric $CO_2$ in (c) is from Köhler et al. (2017)

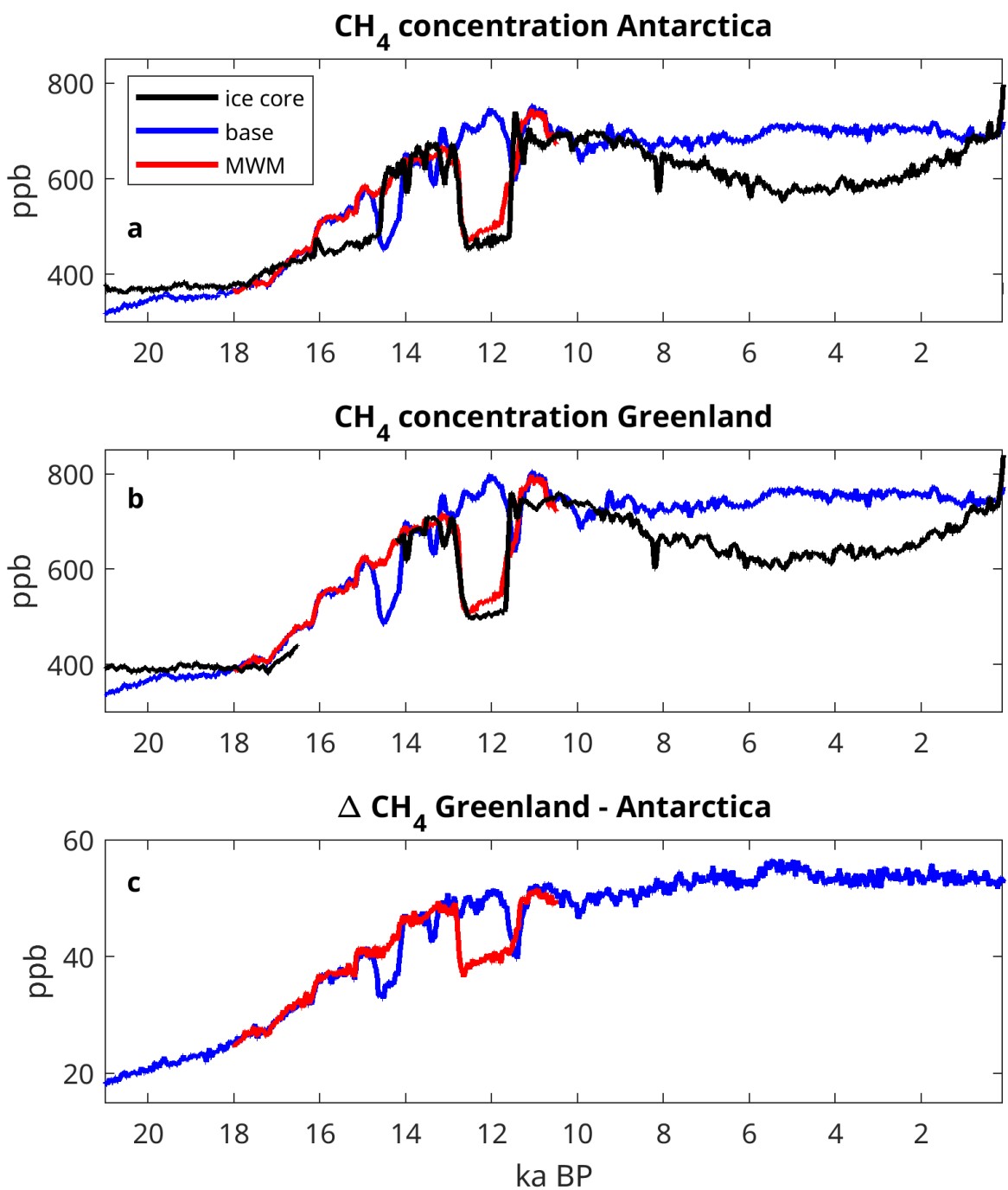

**Figure 2.** Atmospheric concentration of CH$_4$ over Antarctica (a), Greenland (b) in MPI-ESM *base* and *MWM* experiments, as well as ice core data. Antarctic ice core data from Köhler et al. (2017) and Greenland ice core data from Chappellaz et al. (2013) and Beck et al. (2018). (c) interpolar gradient of atmospheric CH$_4$, expressed as difference in CH$_4$ concentration between Greenland and Antarctica.

| Time period | LGM | early BA | late BA | early YD | early Holocene |
|---|---|---|---|---|---|
| | 20 ka | 14.2 ka | 12.9 ka | 12.5 ka | 10.6 ka |
| Antarctica: | | | | | |
| base | 350 | 540 | 660 | 710 | 700 |
| MWM | n/a | 620 | 630 | 480 | 680 |
| ice core | 370 | 620 | 660 | 460 | 690 |
| Greenland: | | | | | |
| base | 370 | 580 | 710 | 760 | 750 |
| MWM | n/a | 663 | 680 | 510 | 730 |
| ice core | 390 | 660 | 700 | 500 | 750 |

**Table 1.** Simulated and reconstructed $CH_4$ concentrations [ppb] at the ice core sites for selected time periods.

At 14.6 ka BP, the transition into the Bølling-Allerød warm period, $CH_4$ increases abruptly by ~150 *ppb*, decreasing again at 12.8 ka BP, the transition into the Younger Dryas. At the end of the YD, at about 11.5 ka BP, $CH_4$ again increases abruptly and stays roughly constant at ~690 *ppb* for the next 2000 yrs. Finally, ice core $CH_4$ slowly decreases by ~120 *ppb* during the early Holocene between 9 ka BP and 4.5 ka BP, recovering after the mid Holocene to ~690 *ppb* at PI.

Modelled atmospheric $CH_4$ over Antarctica is only slightly lower than reconstructed at 20 ka BP (370 ppb ice core, 350 ppb model, Table 1), though the discrepancy is larger at 21 ka BP. Similar to reconstructions, the atmospheric $CH_4$ starts rising significantly at 18 ka BP, but the increase at the beginning of the Bølling-Allerød and the drop in methane during the transition into the Younger Dryas are dissimilar, at least in the *base* experiment: In terms of atmospheric $CH_4$ the Bølling-Allerød seems to start earlier, at 16 ka BP instead of 14.6 ka BP. At 14.38 ka BP, near the beginning of the B-A in reconstructions, the *base* experiment displays a significant drop in atmospheric methane, caused by the AMOC shutdown after MWP 1a. For the transition into the Younger Dryas at 12.8 ka BP, on the other hand, the *base* experiment continues at high levels of atmospheric methane and does not show the decrease in methane displayed by the ice core data as the model shows no climatic transition at this time.

The *MWM* experiment differs from the *base* experiment at the beginning of the B-A, as the atmospheric concentration of methane does not change at the beginning of the Bølling-Allerød. Instead, it is very similar to the ice core data. Furthermore, the atmospheric concentration of methane very rapidly decreases by 25% at the beginning of the Younger Dryas at 12.8 ka BP, similar to ice core data (-29%). At the end of the Younger Dryas, recovery of atmospheric methane occurs slightly earlier than in the ice core data, but not as quickly. The Younger Dryas in our *MWM* experiment, therefore, is quite similar to the ice core data. For atmospheric methane during the Holocene, however, there is a significant divergence between model results and ice-core data: While ice-core $CH_4$ slowly decreases towards the mid-Holocene and increases again during the late Holocene, the model results show constant atmospheric $CH_4$ throughout the Holocene.

A comparison to Greenland ice cores is much more intricate, as recent investigations show excess $CH_4$ in some Greenland ice core records (Lee et al., 2020). This issue has not yet been resolved fully, and we thus do not show a full ice core $CH_4$

timeseries from Greenland. Instead we focus on data derived from the NEEM ice core (Chappellaz et al., 2013), which has been obtained using a continuous-flow measurement technique and is thus presumably less susceptible to the generation of excess $CH_4$, as well as data from the Holocene (Beck et al., 2018), when dust concentration was much lower. For those times when data is available (earlier than 17.5 ka BP, 14 ka BP to PI), the comparison for Greenland (Fig. 2b) is similarly good as the comparison for Antarctica . Thus Greenland $CH_4$ is very similar for 20 ka BP to 18 ka BP, and the evolution of methane during

the Bølling-Allerød - Younger Dryas transition is also very similar in model and ice core data, if one considers the *MWM* experiment (Table 1). Our model thus captures the gradient in methane between Greenland and Antarctica adequately for key periods of the deglaciation. The interpolar gradient, shown here as the difference in atmospheric $CH_4$ concentration between Greenland and Antarctica (Fig. 2c) is positive throughout the experiment, with values at LGM near 20 ppb and PI values of about 55 ppb. Again, the rate of increase is highest between 18 ka BP and 10.5 ka BP, with relatively stable values throughout

the Holocene. Finally, the tropical methane concentration (Fig. A1a) stays in between the values for Antarctica and Greenland throughout the experiment, in contrast to previous studies showing LGM concentrations highest in the tropics (Valdes et al., 2005).

The terrestrial methane fluxes (Fig. 3) largely determine the atmospheric methane concentration. The net $CH_4$ flux (Fig. 3a) increases from $90\,TgCH_4yr^{-1}$ at 20 ka BP to $165\,TgCH_4yr^{-1}$ at PI, with emissions staying very similar between 20 and 18 ka

BP, then increasing more strongly between 18 and 11 ka BP, while staying nearly constant between 10 ka BP and PI. In the *base* experiment, emissions decrease strongly as a response to the AMOC collapse after MWP 1a, with emissions recovering quickly as the AMOC resumes, a pattern continued after MWP 1b. In the *MWM* experiment, however, we see a reduction in net emissions by $32\,TgCH_4yr^{-1}$ in response to the AMOC collapse induced by the meltwater manipulation, followed by a quick recovery as the manipulation ceases. Wetland fluxes (Fig. 3b) are the most important component of the net $CH_4$ flux,

thus their temporal change is rather similar to the net flux, though at a slightly smaller overall magnitude, with wetland emissions of $79\,TgCH_4yr^{-1}$ at 20 ka BP and $142\,TgCH_4yr^{-1}$ at PI, while the emission reduction from the AMOC collapse in the *MWM* experiment is $32\,TgCH_4yr^{-1}$. Here, the AMOC collapse leads to a significant decrease in NH temperatures around the Atlantic ocean, in turn leading to decreased evaporation and thus decreased precipitation, thereby decreasing wetland areas and methane production in the NH tropics. The non-wetland methane fluxes (Fig. 3c) are of significantly smaller magnitude than

the wetland fluxes, with emissions at 20 ka BP of $7.4\,TgCH_4yr^{-1}$ for herbivores, $2.8\,TgCH_4yr^{-1}$ for termites, $3.2\,TgCH_4yr^{-1}$ for fire emissions, and $-2.2\,TgCH_4yr^{-1}$ for the methane uptake in upland soils. Towards the PI state, these fluxes increase to $14.5\,TgCH_4yr^{-1}$, $6.3\,TgCH_4yr^{-1}$, $10.9\,TgCH_4yr^{-1}$ and $-6.3\,TgCH_4yr^{-1}$ for herbivores, termites, fires and uptake, respectively. Here, the differences between the *base* and *MWM* experiments are small, we thus omit the *MWM* experiment from Fig. 3c. The dynamics of the uptake flux appear different from the other fluxes at first glance, with pronounced differences for

Bølling-Allerød and Younger Dryas at the same times as they appear in proxy records. This is due to the fact that the most important driver for the terrestrial methane uptake is the atmospheric concentration of methane (Kleinen et al., 2020), which we prescribed from ice core data for the uptake flux.

We model the atmospheric sink of methane as a function of terrestrial emissions of reactive carbon (RC), as well as $NO_X$ emissions from the soil/vegetation and lightning (Eq. 2). The temporal changes in RC emissions during the deglaciation

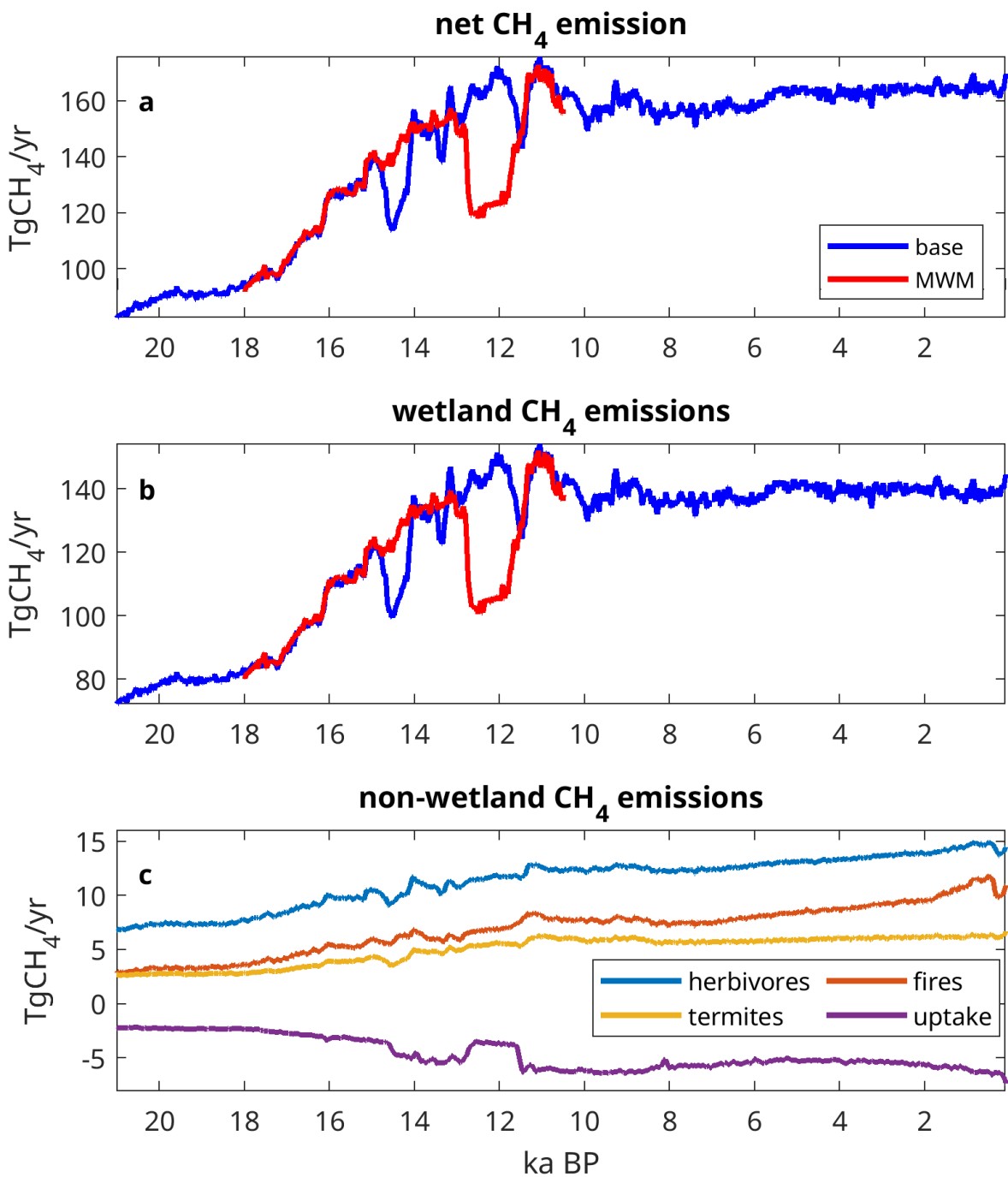

**Figure 3.** Terrestrial methane in MPI-ESM *base* and *MWM* experiments: (a) net flux, (b) wetland emission flux, (c) non-wetland methane emissions from herbivores, fires, termites and terrestrial methane uptake.

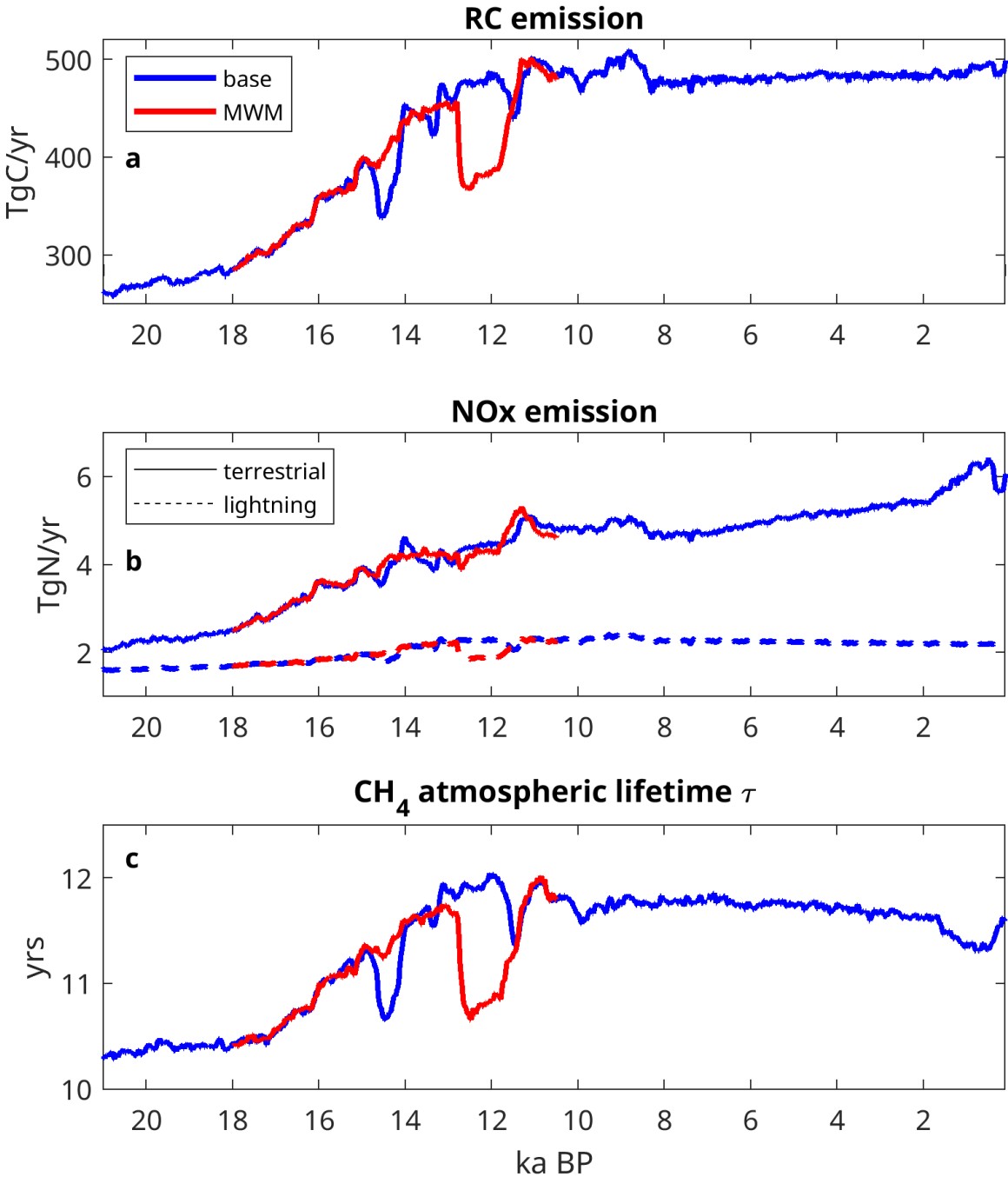

**Figure 4.** Atmospheric sink of CH$_4$: Emissions of reactive carbon (RC) (a), NO$_X$ from soils and lightning (b) and resulting atmospheric lifetime $\tau$ of CH$_4$ (c). $\tau$ increases with increasing RC and decreases with increasing NO$_X$.

(Fig. 4a) are very similar to the changes in $CH_4$ emissions: At 20 ka BP, RC emissions are $270\,TgC\,yr^{-1}$, and at 18 ka BP these start increasing noticeably, reaching a maximum of $500\,TgC\,yr^{-1}$ at 11 ka BP. They are nearly constant, with a very small increasing trend during the Holocene, with values increasing from $477\,TgC\,yr^{-1}$ at 8 ka BP to $490\,TgC\,yr^{-1}$ at PI. The *base* experiment also contains interruptions of the increasing trend after MWP 1a and 1b, which do not occur in the *MWM* experiment. The latter instead displays a significant decrease in RC emissions during the Younger Dryas. Surface $NO_X$ emissions

(Fig. 4b) increase much more gradually than RC emissions, though at a higher relative rate, with total fluxes of $2.2\,TgN\,yr^{-1}$ and $5.8\,TgN\,yr^{-1}$ at 20 ka BP and PI, respectively. They also are much less affected by AMOC fluctuations, thus not having as strong a response to MWPs 1a and 1b in the *base* experiment, or the meltwater-induced AMOC collapse in the *MWM* experiment. As a result, the two experiments do not differ significantly in terms of terrestrial $NO_X$ emissions. This is different for the lightning $NO_X$ emissions ($LNO_X$), which have a minimum of $1.6\,TgN\,yr^{-1}$ at LGM and increase to $2.4\,TgN\,yr^{-1}$ at 9 ka BP,

decreasing thereafter to $2.2\,TgN\,yr^{-1}$ at PI. Decreases after MWPs 1a and 1b are clearly visible in the *base* experiment; in the *MWM* experiment, $LNO_X$ reduces from $2.2\,TgN\,yr^{-1}$ during the BA to $1.9\,TgN\,yr^{-1}$ during the YD.

In both experiments, the simulated atmospheric lifetime of $CH_4$ (Fig. 4c) varies considerably, from $10.4\,yrs$ at 20 ka BP to a maximum of $12\,yrs$ at 12-11 ka BP, slowly decreasing throughout most of the Holocene, from $11.8\,yrs$ at 8 ka BP to $11.6\,yrs$ at 2 ka BP, and a departure to $11.3\,yrs$ at 1 ka BP, before reaching $11.6\,yrs$ at PI. Prior to about 8 ka BP, the variations are

dominated by pronounced changes in RC emissions and $CH_4$ burden. Note that larger RC emissions and $CH_4$ burden increase methane lifetime, whereas larger $NO_X$ emissions have the opposite effect. The general tendency towards higher $NO_X$ emissions is damping the RC-driven increase in methane lifetime. Consequently, the reduction in RC emissions driven by the meltwater events results in a stronger decrease in methane lifetime, since the total $NO_X$ emissions are much less effected by these events. $LNO_X$ is reduced, but soil $NO_X$ is unaffected and even slightly increasing. Across the latitudes, the mean atmospheric lifetime

of $CH_4$ is longest in the SH extratropics, where values range from 19.5 to $22.5\,yrs$ (Fig. A1b), while lifetimes in the NH extratropics are slightly lower, about $1\,yr$ shorter than in the SH. Mean atmospheric lifetimes in the tropics, however, are substantially shorter (Fig. A1c), ranging from 7.2 to $8.3\,yrs$.

## 3.3 The Bølling-Allerød and Younger Dryas

The model experiments we performed do not display the signature in methane concentration expected for the onset of the

Bølling-Allerød, an abrupt increase in atmospheric methane between 14.6 ka BP and 14.5 ka BP. Instead, we see an abrupt increase in atmospheric methane at 16.2 ka BP, 1.6 ka earlier than expected from paleoclimate reconstructions. What is missing in the model experiments, too, is the AMOC signature that would be associated with Heinrich event 1 (H1), a near collapse as recorded in Bermuda rise sediments (McManus et al., 2004; Stanford et al., 2011). Due to the absence of the H1, the Bølling-Allerød thus occurs earlier in our model, and is initially of smaller magnitude.

Comparing net $CH_4$ emissions between the late BA at 13.2 ka BP in the *MWM* experiment and 20 ka BP (Fig. 5), increases in emissions are apparent south of the NH ice sheets, as well as in Siberia. The most prominent increase in emissions, however, occurs in the NH tropics, with increases in Africa and Asia especially prominent. Especially northern Africa is substantially wetter and greener than either at LGM or at present, with the substantial $CH_4$ emissions from the Sahel region very striking.

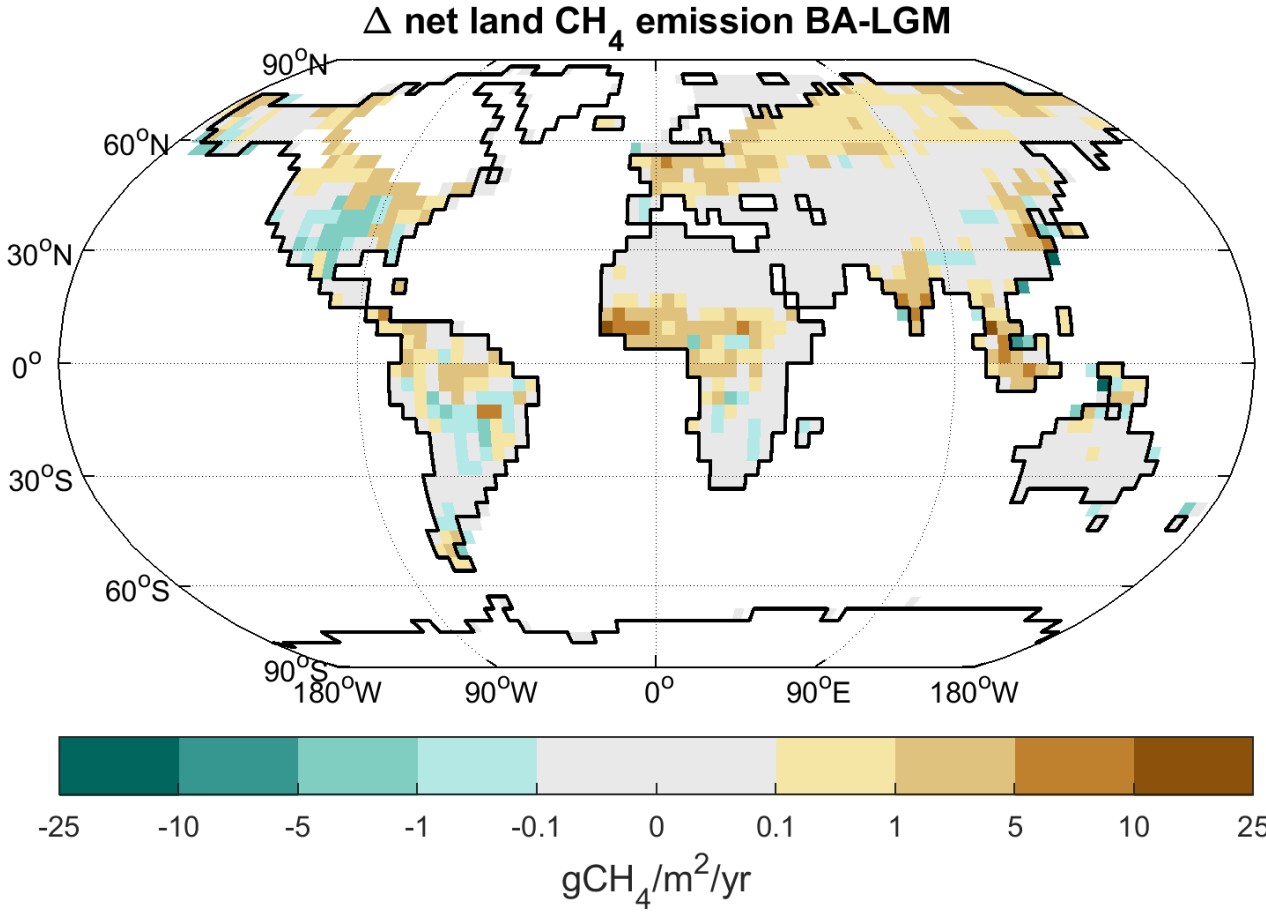

**Figure 5.** Change in net land CH$_4$ emissions between LGM (20 ka BP) and BA (13.2 ka BP) in *MWM* experiment. Shaded areas indicate LGM land points, continental outline is from 13.2 ka BP.

The increase in precipitation here leads to an expansion of wetlands and vegetation, and while the wetland expansion increases the methane emitting area, the expansion of vegetation enhances soil carbon content, thus increasing the substrate available for methane generation.

As the AMOC collapses due to the induced meltwater release at 12.8 ka BP, the region around the North Atlantic Ocean becomes substantially colder, with temperature decreases exceeding $-5K$ in large parts of the Northern and Eastern Atlantic Ocean basin north of the equator. Precipitation in the NH tropics also decreases significantly, with decreases of more than $-1000 mm\,yr^{-1}$ in the tropical Atlantic and precipitation decreases of more than $-400 mm\,yr^{-1}$ in the Sahel region, as well as over India and Indonesia. As a result, methane emissions decrease substantially between the BA at 13.2 ka BP and the YD at 12.5 ka BP (Fig. 6). Most significant are methane emission reductions all over the tropics, largely due to reduced precipitation

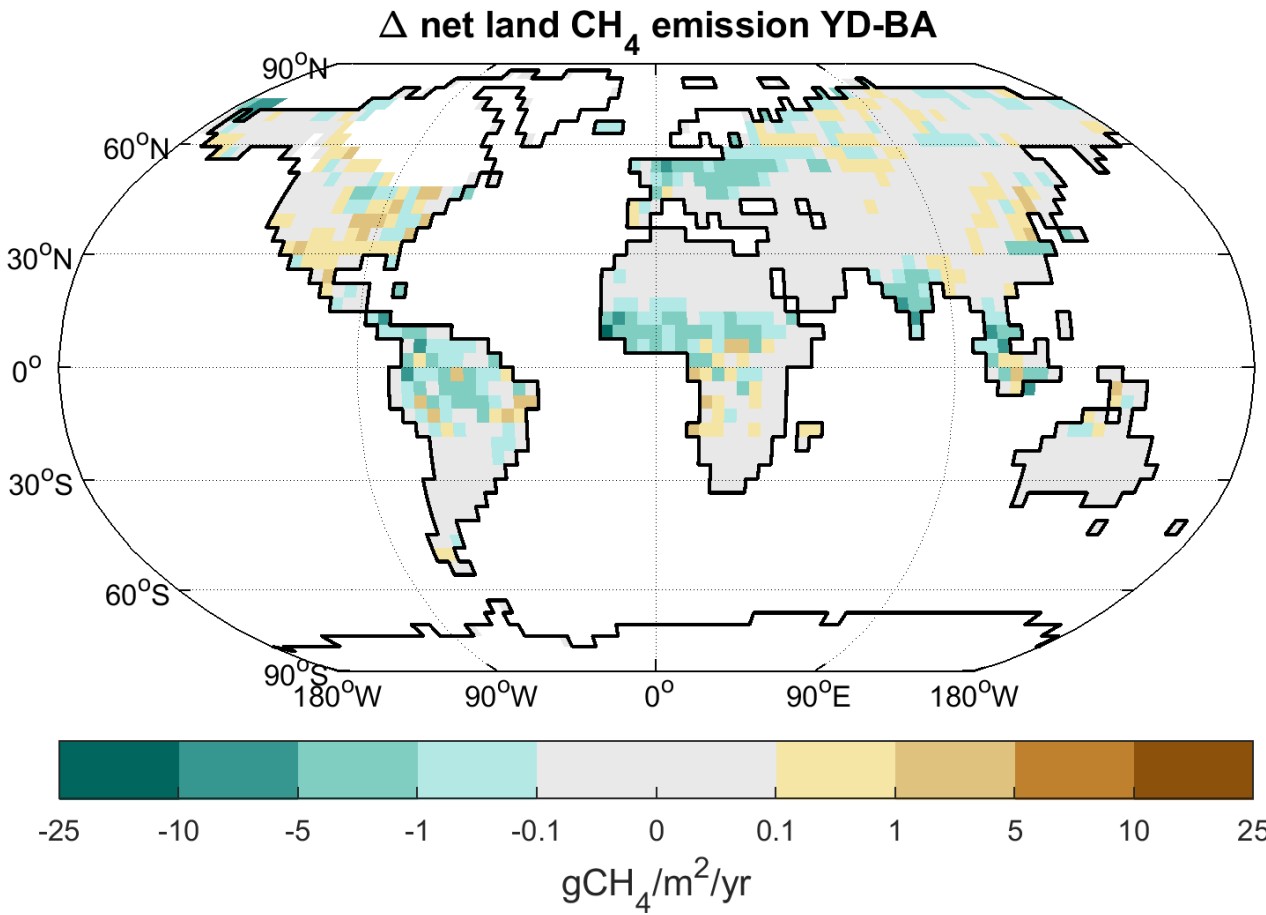

**Figure 6.** Change in net land CH$_4$ emissions between BA (13.2 ka BP) and YD (12.5 ka BP) in *MWM* experiment. Shaded areas indicate BA land points, continental outline is from 12.5 ka BP.

leading to less methane production, but also in Europe, where conditions are substantially colder and dryer than during the BA.

## 3.4 The role of exposed shelf areas

Due to the lower sea level in glacial climate, significant areas of the present-day continental shelf were exposed. In our model, $14.2 \times 10^6 km^2$ of non-glaciated continental shelf that lie below sea level at present were exposed at 20 ka BP. Slightly more than half this area ($7.8 \times 10^6 km^2$) was located in tropical latitudes, predominantly in South-East Asia and Australia. In the NH extratropics, some $5 \times 10^6 km^2$ were exposed, largely in the Laptev Sea and Bering Strait. These areas changed relatively little until 15 ka BP, but started decreasing rapidly afterwards. By 7.8 ka BP most of these areas were flooded, and at 4.3 ka

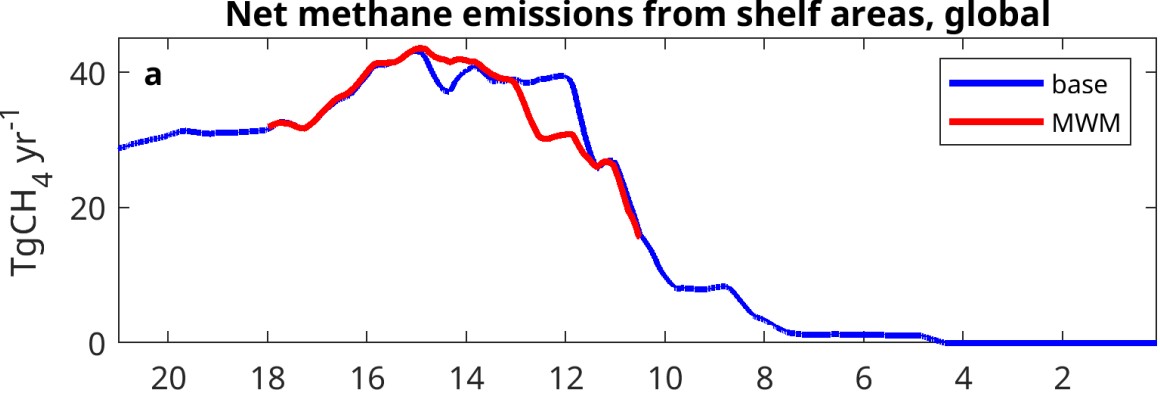

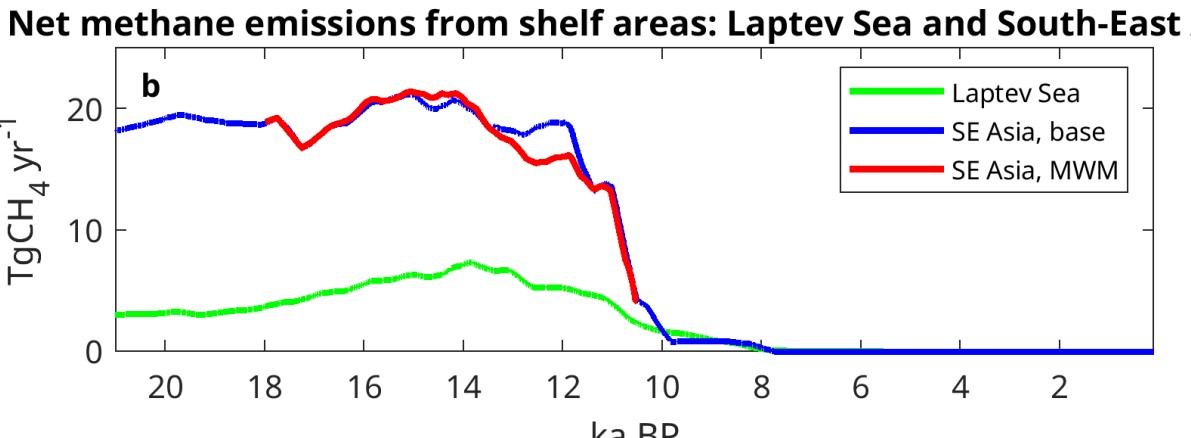

**Figure 7.** Net methane emissions from shelf areas flooded in PI climate. Global emissions (a) and emissions from Laptev Sea and South-East Asia (b).

BP the rising sea level covered the remainder. Our model shows the exposed shelf areas to be a significant source of methane due to significant vegetation growth and wetland formation. With emissions of $31\,TgCH_4\,yr^{-1}$, the global shelf areas emitted a significant fraction of the net methane flux at 20 ka BP (Fig. 7a). Due to global warming, the shelf emissions increased further to $40\,TgCH_4\,yr^{-1}$ at 15 ka BP, but declining subsequently as sea level rise started to submerge these areas. The bulk of this flux

$-\ 20\,TgCH_4\,yr^{-1}$ at 20 ka BP – is emitted from the extensive shelf areas in South-East Asia (Sunda shelf) and north-western Australia (Sahul shelf), combined in Fig. 7b as South East Asia. Here, emissions increased slightly between 20 ka BP and 14 ka BP, subsequently dropping to near zero by 10 ka BP. While the AMOC collapse in the *MWM* experiment does affect the region, decreasing emissions slightly, the effect is less pronounced here than in other regions. In non-tropical regions, the largest exposed shelf area in the Laptev Sea and Bering Strait had emissions of some $3.2\,TgCH_4\,yr^{-1}$ at 20 ka BP, which rose

to $6.9\,TgCH_4\,yr^{-1}$ at 14 ka BP, falling subsequently to zero at 8 ka BP when the remaining shelf area was flooded.

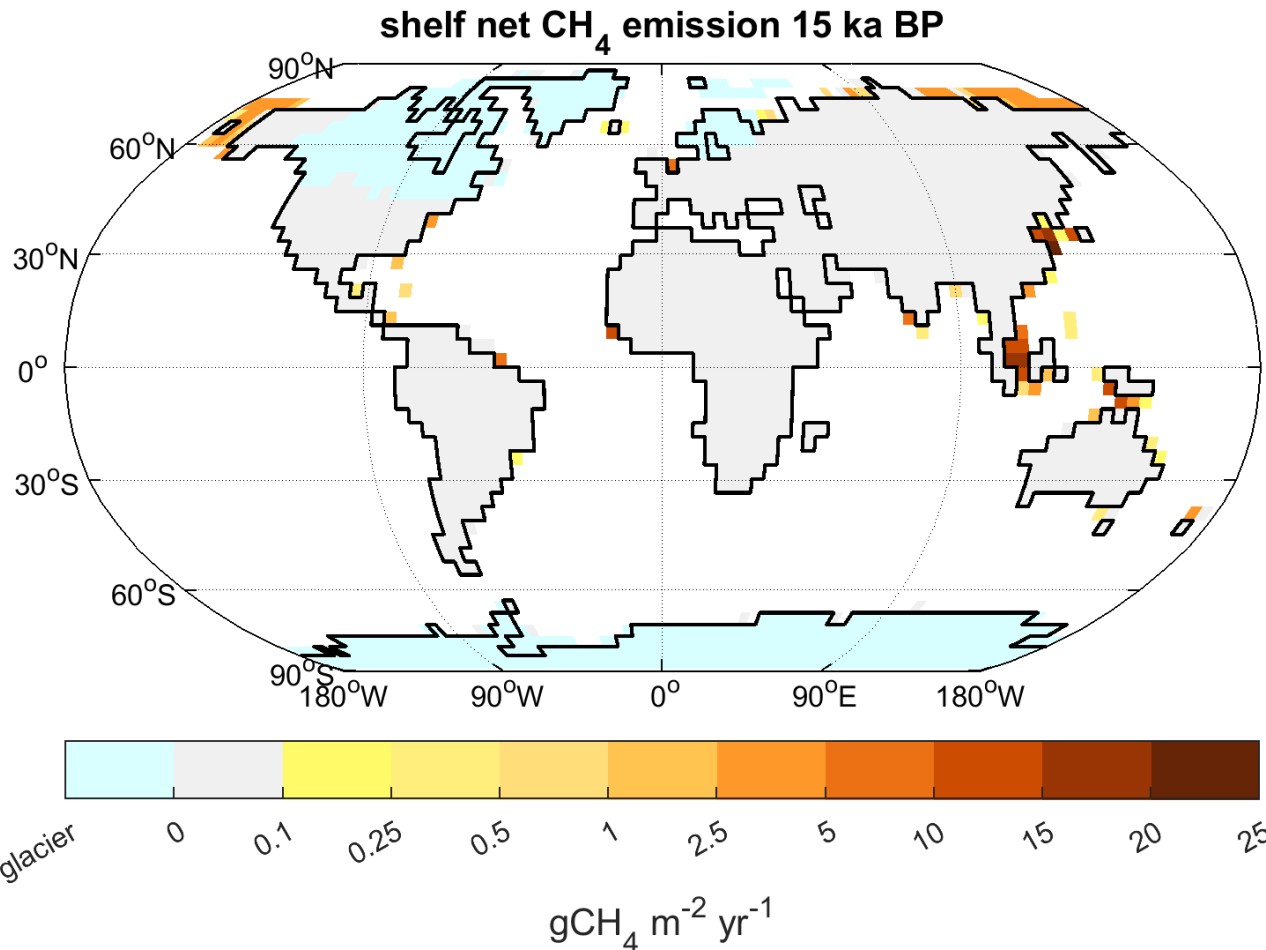

**Figure 8.** Net shelf CH4 emissions at 15 ka BP, the emission maximum. Shaded areas indicate land points at 15 ka BP, continental outline is from the PI state.

There were further regions with some shelf methane emissions, but the aforementioned areas cover the most important source regions at 15 ka BP (Fig. 8). However, the highest net emissions originate in a grid point in the East China Sea, just east of present-day Shanghai. The bathymetric data used to determine wetland area on the shelves indicates a very flat area at this location, which results in a grid-cell mean inundation of nearly 25% at 15 ka BP. Whether this high value accurately reflects conditions at the time when the shelf was exposed, or whether it is due to subsequent changes in bathymetry caused by the Yangtze river sediment load is an open question we are not qualified to answer.

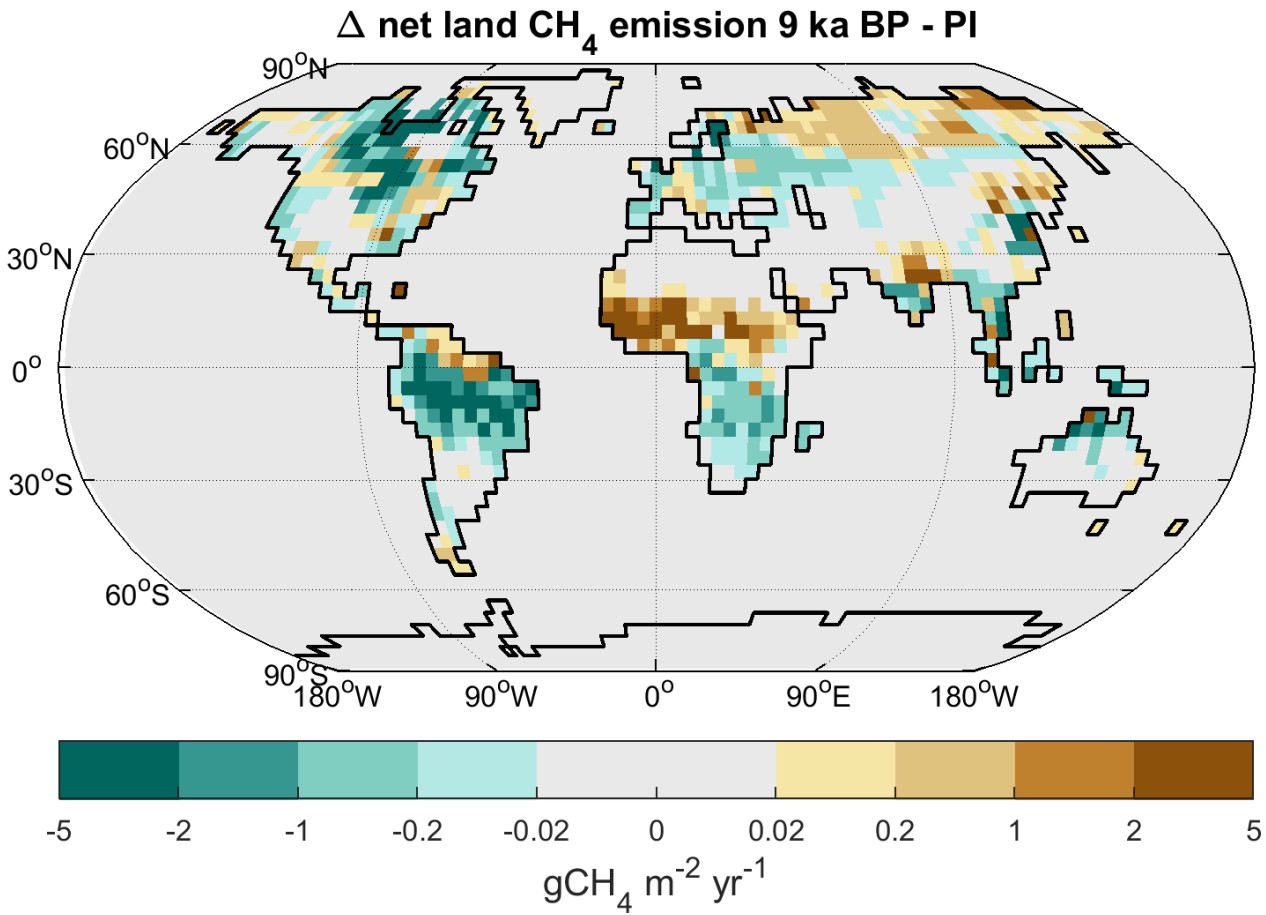

**Figure 9.** Net CH$_4$ emission change 9 ka BP - PI, continental outline is from 9 ka BP.

## 3.5 Regional distribution of methane fluxes over time

During the early Holocene, at 9 ka BP, the total methane emissions were very similar to the PI state. However, the spatial distribution differed significantly (Fig. 9). Due to the recent deglaciation of North America, with some parts of the Laurentide ice sheet still remaining, CH$_4$ emissions from North America were strongly reduced, while emissions from northern Siberia were enhanced in comparison to PI, due to enhanced vegetation growth (and thus soil carbon) from increased solar radiation during the boreal summer season. Furthermore, emissions from South America were reduced. The most striking difference to the PI state, though, were the changes in NH tropical Africa. At present, northern Africa is not an important source of methane. Large parts of the continent are rather dry, thus not producing significant amounts of methane. During the early Holocene, however, the North African monsoon was considerably stronger, leading to significantly enhanced precipitation across the

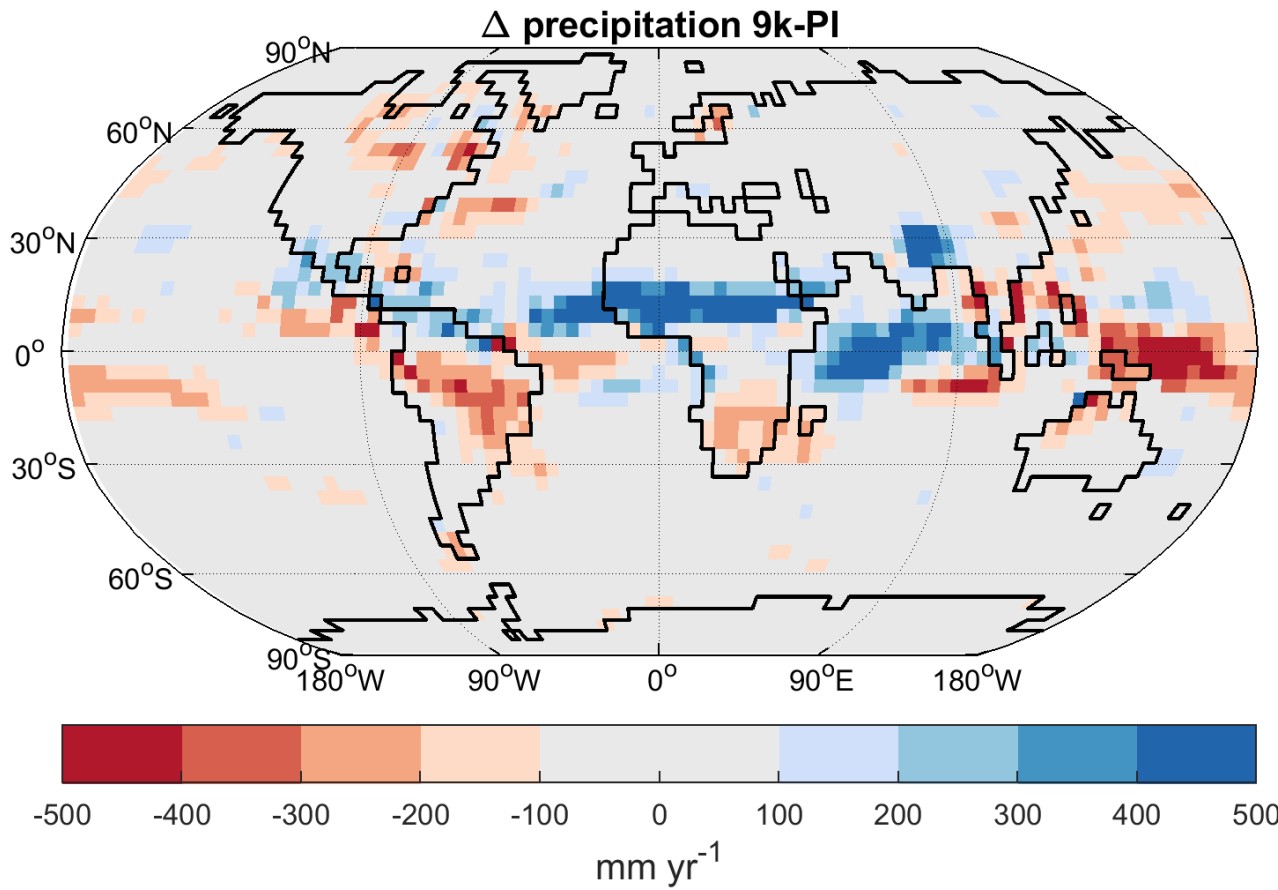

**Figure 10.** Precipitation change 9 ka BP - PI, continental outline is from 9 ka BP.

Sahel region, reaching into the southern Sahara (Fig. 10) (Dallmeyer et al., 2020, 2021). As a result, vegetation and wetlands in the Sahel expanded and methane emissions were substantially stronger than at PI.

Looking at the timeseries of the regional distribution of methane fluxes since the LGM in our experiments, the SH tropics had the largest net emissions at LGM, with NH tropics and extratropics each having slightly lower emissions. As emissions increased during the deglaciation, their share in the NH tropics and extratropic increased faster than that from the SH tropics (Fig. 11a). Emissions in the NH extratropics increased markedly between 18 ka BP and 11 ka BP, rising from $33\,Tg\,CH_4\,yr^{-1}$ to $60\,Tg\,CH_4\,yr^{-1}$ 7000 years later, with little change afterwards. Emissions in the NH tropics, on the other hand, were strongly affected by the AMOC perturbation events, after MWP 1a in the *base* experiment and at 12.8 ka BP in the *MWM* experiment. Here, emissions also started to increase from $30\,Tg\,CH_4\,yr^{-1}$ at 18 ka BP, and they continued to rise until reaching $62\,Tg\,CH_4\,yr^{-1}$ at 9 ka BP. In the *base* experiment, this increase was interrupted after MWP 1a, here $CH_4$ emissions dropped

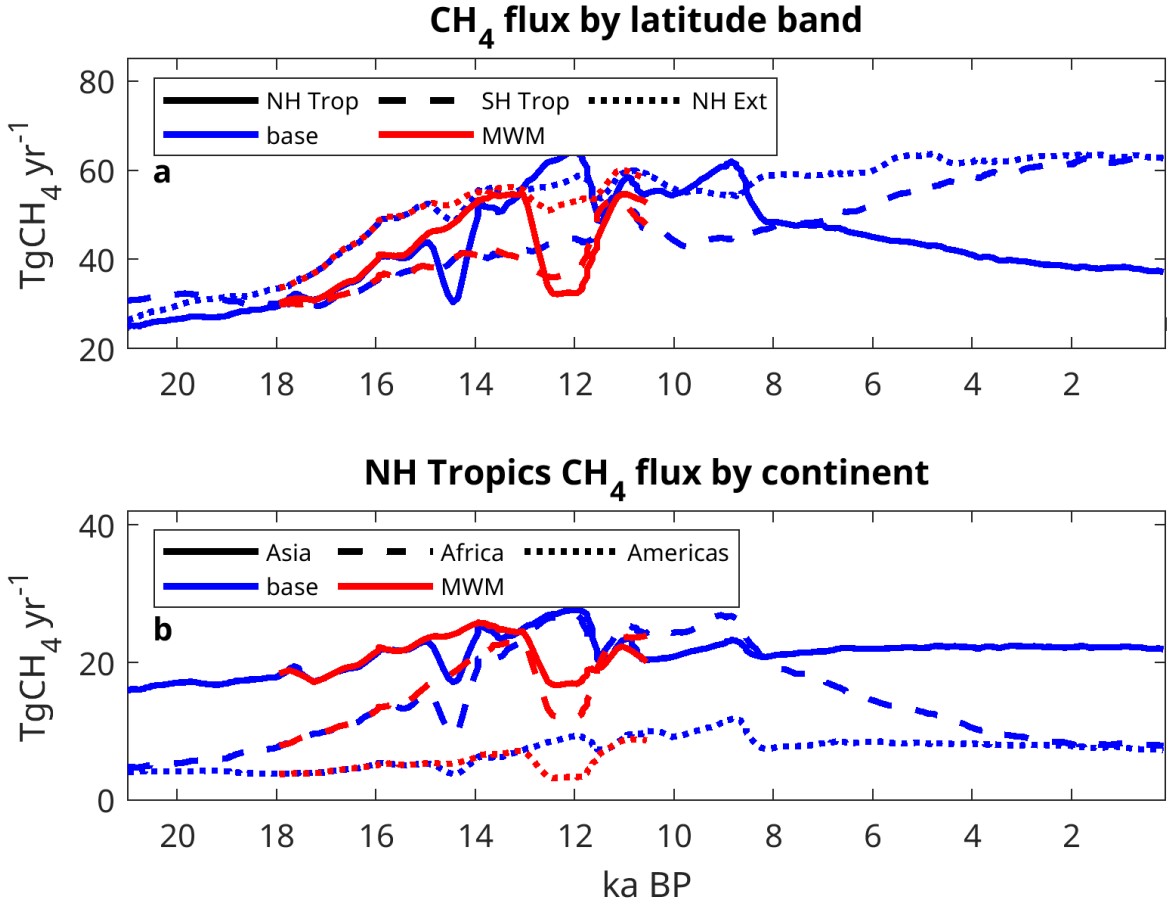

**Figure 11.** Net $CH_4$ flux over time by latitude band (a), and by continent (b) for the NH tropics (0-30°N).

from 46 to $27\,TgCH_4\,yr^{-1}$ during the period from 15 to 14.4 ka BP. In the *MWM* experiment, on the other hand, emissions in the NH tropics strongly reacted to the imposed Younger Dryas event, with emissions dropping from 55 to $31\,TgCH_4\,yr^{-1}$ between 12.8 and 12.5 ka BP. Investigating the changes in the NH tropics by continent (Fig. 11b), it is clear that the bulk of the long-term changes in methane fluxes from this region are from NH tropical Africa, with African $CH_4$ emissions increasing

370 from $5\,TgCH_4\,yr^{-1}$ at 20 ka BP to some $26\,TgCH_4\,yr^{-1}$ between 13 and 8.5 ka BP, with emissions subsequently decreasing gradually to $8\,TgCH_4\,yr^{-1}$ after 2 ka BP. In comparison to these fluxes from NH Africa, fluxes from NH tropical Asia and America change very little, except for the rapid changes in fluxes from NH tropical Asia connected to AMOC changes after MWP1a in the *base* and during the Younger Dryas in the *MWM* experiment.

Finally, focusing on the Holocene changes after 8 ka BP, we see a decreasing trend in emissions from the NH tropics

375 (Fig. 11a), while emissions from the SH tropics increase. The latter increase is due to a generally increasing trend in precipitation over South America (Fig. 10), leading to increasing emissions from the Amazon region, while the former is due to a

decreasing trend in precipitation over NH tropical regions, mainly in Africa, but also over northern India. These precipitation trends are ultimately caused by orbital changes with the change in precession leading to a decrease in radiation reaching NH tropical areas and an increase in radiation reaching SH tropical areas, causing changes in land-sea temperature contrasts and thus changes in monsoon circulation and related precipitation (Dallmeyer et al., 2021). We thus see a very similar trend to Singarayer et al. (2011), who saw decreasing emissions from the NH tropics and increasing emissions from the SH over the last 8 ka. Singarayer et al. (2011) were thus able to explain the Holocene trend in reconstructed $CH_4$, a decrease from early to mid Holocene, followed by an increase from mid to late Holocene. In our model, however, these trends more or less cancel each other, thus leading to very small changes in overall methane emissions, despite the substantial changes in regional fluxes.

## 4 Discussion

The evolution of climate in our transient deglaciation experiment generally seems to be similar to climate reconstructions, with the reconstruction by Shakun et al. (2012) showing smaller changes in global mean temperature and the reconstruction by Osman et al. (2021) showing larger changes. Regional details, though, may not necessarily agree with reconstructions. Examples here are the freshwater flux from MWP 1a leading to a collapse of the AMOC in the *base* experiment, or the missing H1 Heinrich event leading to an earlier warming related to the B-A onset in the *base* and *MWM* experiments. One could argue that a model configuration with prescribed ice sheets and meltwater fluxes, as used here, is not capable of simulating a full Heinrich event, as a true H event is a coupled mode of ice-sheet and ocean dynamics (Ziemen et al., 2019) that requires an interactive ice sheet model to be captured completely. Nonetheless our model experiment contains two events that show most of the climatic characteristics of a Heinrich event, the AMOC collapse after MWP 1a in the *base* experiment and the induced transition into the Younger Dryas in the *MWM* experiment. The latter event also shows the methane response one would expect from a Heinrich event (Fig. 6): A general decrease in circum-Atlantic methane emissions, possibly extending to further tropical methane emission areas. This picture is, however, strongly dependent on the background state: As emissions from Europe are smaller under full glacial conditions than under B-A conditions, a Heinrich event under full glacial conditions would lead to a much smaller decrease in emissions than during the Younger Dryas.

Balancing the methane budget over the course of the deglaciation has proven to be very challenging. Flux estimates for some of the methane source fluxes vary widely in the literature and may be quite different from the ones we obtain. Bock et al. (2017), for example, estimate from $CH_4$ stable isotopes that the combined flux from biomass burning and geological sources is some $90\,TgCH_4\,yr^{-1}$ in interglacials and some $70\,TgCH_4\,yr^{-1}$ in glacials. Along similar lines, Saunois et al. (2020), based on Etiope and Schwietzke (2019), estimate some $45\,TgCH_4\,yr^{-1}$ for present-day geological emissions. Taking the top-down estimate of total natural methane fluxes for the present-day from Saunois et al. (2020), $232\,TgCH_4\,yr^{-1}$, the Bock et al. (2017) estimate, taken at face value, would imply that about 39% of the present-day natural $CH_4$ emissions would be from wildfires and geological sources, indicating that a substantial reduction in all other natural sources of methane would be required in order to balance the global $CH_4$ budget. Our model shows some $165\,TgCH_4\,yr^{-1}$ and $90\,TgCH_4\,yr^{-1}$ for total net emissions at PI and LGM, respectively, and the Bock et al. (2017) estimate would thus imply that 54% of preindustrial emissions and

about 78% of the LGM net methane emissions are from biomass burning and geological sources. Furthermore, the estimates from Saunois et al. (2020) would imply that roughly half of the total LGM methane emissions are of geological origin, as we are not aware of any mechanisms that would lead to lower geological fluxes at LGM in comparison to the present. The only way to reconcile these flux estimates with reconstructed atmospheric $CH_4$ concentrations would be to assume that either the atmospheric lifetime of $CH_4$ was substantially lower at LGM than at present, which goes against the present understanding of methane chemistry (Murray et al., 2014), or to assume that the other natural sources of methane, especially wetlands which are generally acknowledged to be the largest natural source of $CH_4$ (Saunois et al., 2020), would be drastically reduced in comparison to our experiments and other estimates. We therefore assume that these flux estimates are an overestimate.

Our present publication generally confirms the results from time-slice experiments we obtained earlier (Kleinen et al., 2020), however importantly integrating a methane sink component into the modelling system and enabling the investigation of highly transient changes in methane as during the BA-YD transition. Thus we corroborate that the large-scale change in atmospheric methane from LGM to Holocene is mainly due to changes in wetland emissions, with the tropical areas being the main emitting region, while NH extratropics play a secondary role. This change in wetland emissions can be attributed to increases in soil carbon storage, increases in atmospheric $CO_2$ and further climate changes, with warming playing the most prominent role (Kleinen et al., 2020). While there is a number of uncertain parameters in the wetland methane emission model, changes in these tend to have very similar effects in both LGM and PI climate states, proportionally adjusting the emission strengths so that their LGM to PI ratio remains little affected. The latter is instead determined by the changes in soil C, atmospheric $CO_2$ and climate. Therefore factors like $CO_2$ fertilisation, the increase in vegetation productivity with increasing atmospheric $CO_2$, would need to be adjusted in order to affect the ratio of LGM to PI $CH_4$ emissions. As the difference in terrestrial carbon storage between LGM and PI is on the high side (Jeltsch-Thömmes et al. (2019), for example, estimate 450 to 1250 PgC for the LGM to PI change in terrestrial C, including substantial C stores like peatlands which we do not consider in our model), a decrease in $CO_2$ sensitivity might improve overall results.

We calibrated the contributions by the different methane sources to the total flux to be conformal to the present-day source distribution (Saunois et al., 2016, 2020), and the relative contributions of the single sources to the total emissions change very little over the course of the deglaciation. In terms of total emissions, Valdes et al. (2005) assume LGM emissions of $152.4\,TgCH_4\,yr^{-1}$ and PI emissions of $198.9\,TgCH_4\,yr^{-1}$, while Hopcroft et al. (2017) obtain $129.7\,TgCH_4\,yr^{-1}$ at LGM and $197.2\,TgCH_4\,yr^{-1}$ at PI. Both of these estimates are higher than our results (net emissions of $90\,TgCH_4yr^{-1}$ at 20 ka BP and $165\,TgCH_4yr^{-1}$ at PI), requiring a shorter atmospheric lifetime of $CH_4$ than we obtain.

Both Valdes et al. (2005) and Hopcroft et al. (2017) assume oceanic $CH_4$ emissions, with LGM emissions of the order of $11\,TgCH_4yr^{-1}$ and PI emissions of the order of $14\,TgCH_4yr^{-1}$. We only consider oceanic emissions from geological sources (Etiope, 2015; Saunois et al., 2016, 2020), neglecting biogenic oceanic sources assumed to be small. For the geological emissions, the total amount is highly debated at present, with direct measurements showing substantially higher estimates (Etiope, 2015; Mazzini et al., 2021) than ice-core based reconstructions of emissions during the YD (Petrenko et al., 2017) or PI (Hmiel et al., 2020). We chose geological emissions of $5\,TgCH_4yr^{-1}$ at the upper limit still compatible with ice-core measurements. Assuming geological emissions of $45\,TgCH_4yr^{-1}$, more in line with Saunois et al. (2020), would make it extremely difficult to

match the LGM methane budget, as the fraction of geological sources would then make up roughly half of the total emissions, implying even larger reduction in all other methane fluxes for the LGM in comparison to PI than already required.

The modelled wetland emission fluxes are dependent on inundated areas extent, as well as on soil carbon content within those inundated areas. Our model overestimates inundation in tropical areas (Kleinen et al., 2020) in comparison to some remote-sensing estimates (Prigent et al., 2012). The latter, however, is based on a combination of optical and radar sensor data and likely susceptible to underestimation of inundation in areas with dense forest canopy, such as tropical rainforests (Melton et al., 2013). Unfortunately, due to lack of other reference data, reconciling the model and remote-sensing estimates is not possible yet. If we assume that the overestimate of tropical wetland areas was significant, though, it would decrease tropical emissions, adding more weight to the extratropical areas. However, the modelled wetland methane emission fluxes for PI and PD climate states (Kleinen et al., 2020, 2021) fall well within the range of wetland fluxes shown in the latest *Global Carbon project Methane assessment* (Saunois et al., 2020), and the latitudinal distribution of wetland methane emissions is similar to assessments in atmospheric inversion studies (Bousquet et al., 2011), where tropical areas are the source for 62-77% of global wetland emissions. Thus we conclude that our model generally captures the wetland emissions correctly.

Methane emissions from wildfires range from $3.2\,TgCH_4\,yr^{-1}$ at LGM to $10.9\,TgCH_4\,yr^{-1}$ at PI in our experiments, thus substantially lower than the Bock et al. (2017) estimate for combined emissions from wildfires and geological sources. However, some evidence for past wildfires during recent millennia also points towards higher fire occurrence than shown by our model (Marlon et al., 2008; Nicewonger et al., 2020). The SPITFIRE fire model we use (Lasslop et al., 2014) requires two factors for a fire to occur: a sufficient amount of fuel and an ignition source. The former is a function of climate (biomass, precipitation and temperature), while the latter is modulated by either lightning intensity or human activities parameterised from the population density. With less carbon on land (Fig. 1c) at LGM, as well as a much less dense human population, igniting presumably much fewer fires, missing wildfire sources would be attributed to underestimated lightning-induced emissions in glacial climate. However, the lightning model we use for the atmospheric chemistry (Price and Rind, 1992, 1993) indicates the decrease in lightning in colder climates due to reduced convective activity (see Lightning $NO_X$ in Fig. 4b for comparison), thus rendering higher fire emissions in glacial climate less likely. Paleorecords covering the last one (Marlon et al., 2008) to two millennia (Nicewonger et al., 2020) indicate a higher fire activity than shown by our model, possibly at levels similar to present day. However, longer-term studies based on charcoal records from the LGM to PI (Power et al., 2008) show an overall increase in fire occurrence over all continents as the deglaciation progresses, qualitatively very similar to fire occurrence in MPIESM in our study.

For methane emissions from herbivorous mammals, this source category is at present dominated by domesticated animals, mainly cows, and both the densities and the species distributions of wild herbivorous mammals in an Earth System untouched by humans, as we assume it to be for the LGM state, is completely unknown. We thus tied the emissions by herbivorous mammals to the net primary production as a proxy for food availability. Previous estimates (Crutzen et al., 1986; Chappellaz et al., 1993) were derived using a different methodology, estimating key species populations and extrapolating emissions from these. Chappellaz et al. (1993) thus estimate herbivore emissions of $15\,TgCH_4\,yr^{-1}$ for the Preindustrial Holocene and of $20\,TgCH_4\,yr^{-1}$ for the LGM, arguing that grasslands expanded by 50% at LGM, thus enlarging the habitat. In our model

experiments, however, we find that grasslands do not expand at the LGM in comparison to PI. Instead, grasslands are slightly larger in the PI state than at LGM. It may be that the discrepancy in areal estimates is due to Chappellaz et al. (1993) assuming grassland in LGM eastern Siberia, but our model instead finds that polar deserts expand in there due to extremely dry conditions. Other estimates of herbivore emissions generally go back to the estimate by Crutzen et al. (1986), which also forms the basis for the Chappellaz et al. (1993) estimate, and thus do not need to be discussed here further. Smith et al. (2016), however,

also estimated Late Pleistocene and Preindustrial methane production form herbivorous mammals. They estimate herbivore emissions of $150\,Tg\,CH_4\,yr^{-1}$ for the late Pleistocene, which is slightly less than double our estimate of net total emissions at LGM. Thus their estimate would require a dramatically shorter lifetime for atmospheric $CH_4$ than for the PI state, which is considered incompatible with present-day understanding of the methane cycle (Murray et al., 2014; Hopcroft et al., 2017).

The soil uptake of methane is calculated from the diffusion of methane into dry soils, where it is subsequently oxidised. The

magnitude of this flux is determined by the rate of diffusion into the soil, as well as the oxidation rate of methane in the soil. Here, we do not consider the potential enhancement of high latitude methane uptake by high-affinity methanotrophs suggested by Oh et al. (2020). In our model, the rate of methane uptake is primarily limited by the rate of diffusion of $CH_4$ and $O_2$ into the soil, while the oxidation rate is of secondary importance. An increase in the latter in high latitudes would thus have a small impact on total uptake. Depending on its formulation and parameterisation, in other models the mechanism by Oh et al. (2020)

might have a greater impact. We also prescribed atmospheric methane concentrations from ice core data (Köhler et al., 2017) when determining the uptake flux. Obviously there is a small discrepancy in comparison to the flux we would have obtained if we had used the modelled methane concentration. However, the impact of this discrepancy on the atmospheric methane concentration is small, as modelled and ice core $CH_4$ are very similar to each other for most of the experiment. One exception to this is the mid-Holocene period, when the modelled $CH_4$ is substantially higher than the reconstruction. Thus, the methane

uptake during this period would be higher than shown here, but not high enough to significantly reduce the $CH_4$ concentration.

The atmospheric sink of methane in our model is simulated using the total reactivity fields obtained from the EMAC model (Joeckel et al., 2010) modulated according to climate changes as shown in Eq. 2: atmospheric $CH_4$ removal increases (equivalently decreasing the lifetime) with increasing $NO_X$ emission from soils and lightning and reduction in RC emissions. While methane sinks obtained in the EMAC model allow re-combining them in the total sink fields (e.g. for isotope-enabled studies),

in the current MPIESM experiments we have used the aggregated sink formulation which does not allow determining changes in the contributions from the different components over the course of the deglaciation. The reason for this more pragmatic implementation, in addition to optimisation of model simulation resources, is that we attribute most of the variation in the $CH_4$ sink to changes in tropospheric OH abundance, driven in turn by changes in emissions of $NO_X$ and RC. Changes to the atmospheric Cl-initiated sink of methane may be important for estimating changes in methane isotope ratios since PI (Levine

et al., 2011), however, in pre-industrial conditions in EMAC it accounts for less than 0.1% of the total methane sink, making it substantially less important for the evolution of pre-industrial $CH_4$.

Finally, the mid-Holocene decrease in atmospheric methane is the one aspect of the development of atmospheric methane in the time from LGM to PI that we were unable to reproduce in a satisfactory way. Singarayer et al. (2011) were able to attribute these changes to the orbital forcing. They found that due to precession-caused insolation changes, wetland emissions

from the SH tropics increase after 5 ka BP, while the emissions from the NH tropics decrease, with the NH decrease being smaller than the SH increase after 5 ka BP. We see a very similar behaviour in our model: wetland emissions from the NH tropics, especially from Africa, decrease after 8 ka BP, while emissions from the SH tropics, predominantly the Amazon region, increase (Fig. 11). The net result in our case, though, is that the NH decrease is exactly compensated by the SH increase, thus leading to no change in total emissions. Our model thus produces constant emissions from 8 ka BP to PI, leading to a constant

atmospheric concentration with neither a decrease before 5 ka BP nor an increase after 5 ka BP being apparent. We suspect that this behaviour may be due either to NH extratropical emissions or to the monsoon changes in north Africa. The total NH extratropical emissions are rather stable over the course of the Holocene (Fig. 11), despite large regional changes (Fig. 9), as changes in one region are compensated by changes in other regions. On the other hand, the expansion of the African monsoon system in our model is relatively limited in comparison to reconstructions. The latter show an expansion of the monsoon into

the Sahara, while we only get an increase in the Sahel. The larger extent in the reconstructions would imply a different temporal behaviour with a narrower emission peak. With this change in African emissions, the emission decrease from the NH tropics would be less than the emission increase from the SH tropics, thus leading to the methane trajectory observed in ice cores.

## 5   Conclusions

Our model experiments demonstrate – for the first time – how the complete methane cycle changes over the course of the

deglaciation. We found that the atmospheric lifetime of $CH_4$ has increased slightly, from $10.4\,yrs$ at 18 ka BP to a maximum of $12\,yrs$ at 12 ka BP, implying that the observed doubling of the atmospheric $CH_4$ concentration during this interval has to be explained primarily by changes in $CH_4$ emissions. The model is capable of simulating such changes in $CH_4$ sources, with wetland emissions increasing from $80\,TgCH_4\,yr^{-1}$ to $150\,TgCH_4\,yr^{-1}$ primarily driven by increases in vegetation productivity. We reproduce all major features of the deglacial ice-core methane record, with the exception of the observed mid-Holocene

minimum in methane, as a decrease in simulated NH methane sources is perfectly balanced by an increase in SH sources, leading to almost no change in the $CH_4$ concentration. For much of the deglaciation, our atmospheric transport model reproduces both Antarctic and Greenland methane records, thus also capturing the interhemispheric gradient.

We also are able to simulate significant emissions of $CH_4$ from shelf areas which were flooded in the course of the deglaciation. Simulated changes in total terrestrial carbon storage of ca. 720 PgC increase from LGM to PI are at the upper end of

modelling estimates.

Some of the abrupt deglacial methane changes, however, cannot be reproduced spontaneously, but rather require dedicated model setups: As shown in our *base* experiment (and by Kapsch et al. (2022)), an application of meltwater forcing from ice sheet reconstructions will not lead to climate changes as reconstructed from proxy data. The meltwater input from MWP 1a leads to a collapse of the North Atlantic AMOC circulation and a strong cooling event in circum-Atlantic areas at the time

when the Bølling-Allerød warming would rather be expected. With appropriate control of the ice sheet meltwater, however, we were able to reproduce the entire deglaciation sequence in experiment *MWM*. Thus storing the meltwater from the Laurentide ice sheet and releasing the accumulated meltwater at 12.8 ka BP leads to an sequence of Bølling-Allerød warming, Younger

Dryas cooling, and Preboreal warming that is, in terms of atmospheric methane, very close to ice core records from Antarctica and Greenland.

*Code and data availability.* The primary data, i.e. the model code for MPI-ESM, are freely available to the scientific community and can be accessed with a license on the MPI-M model distribution website http://www.mpimet.mpg.de/en/science/models (last access: 29 September 2022). In addition, secondary data and scripts that may be useful in reproducing the authors' work are archived by the Max Planck Institute for Meteorology. They can be obtained by contacting the first author or publications@mpimet.mpg.de.

The full model output is available from the DKRZ Earth System Grid node at https://doi.org/10.26050/WDCC/PMMXMCHTD (Kleinen et al., 2023a), and aggregated methane timeseries are available at https://doi.org/10.5281/zenodo.7670389 (Kleinen et al., 2023b).

## Appendix A: Additional Figures

*Author contributions.* TK: Model development, experiment design and analysis, main author of text. SG: Atmospheric sink development, interpretation of results. BS: Atmospheric sink, interpretation of results. VB: Experiment interpretation. All: Discussion of results, editing of manuscript.

*Competing interests.* None.

*Acknowledgements.* We thank two anonymous reviewers for their valuable constructive criticism. We also thank Anne Dallmeyer for comments on an earlier version of this manuscript. We acknowledge support through the project Palmod, funded by the German Federal Ministry of Education and Research (BMBF), Grant Nos. 01LP1921A and 01LP1921B. This work used resources of the Deutsches Klimarechenzentrum (DKRZ) granted by its Scientific Steering Committee (WLA) under Project ID bm1030.

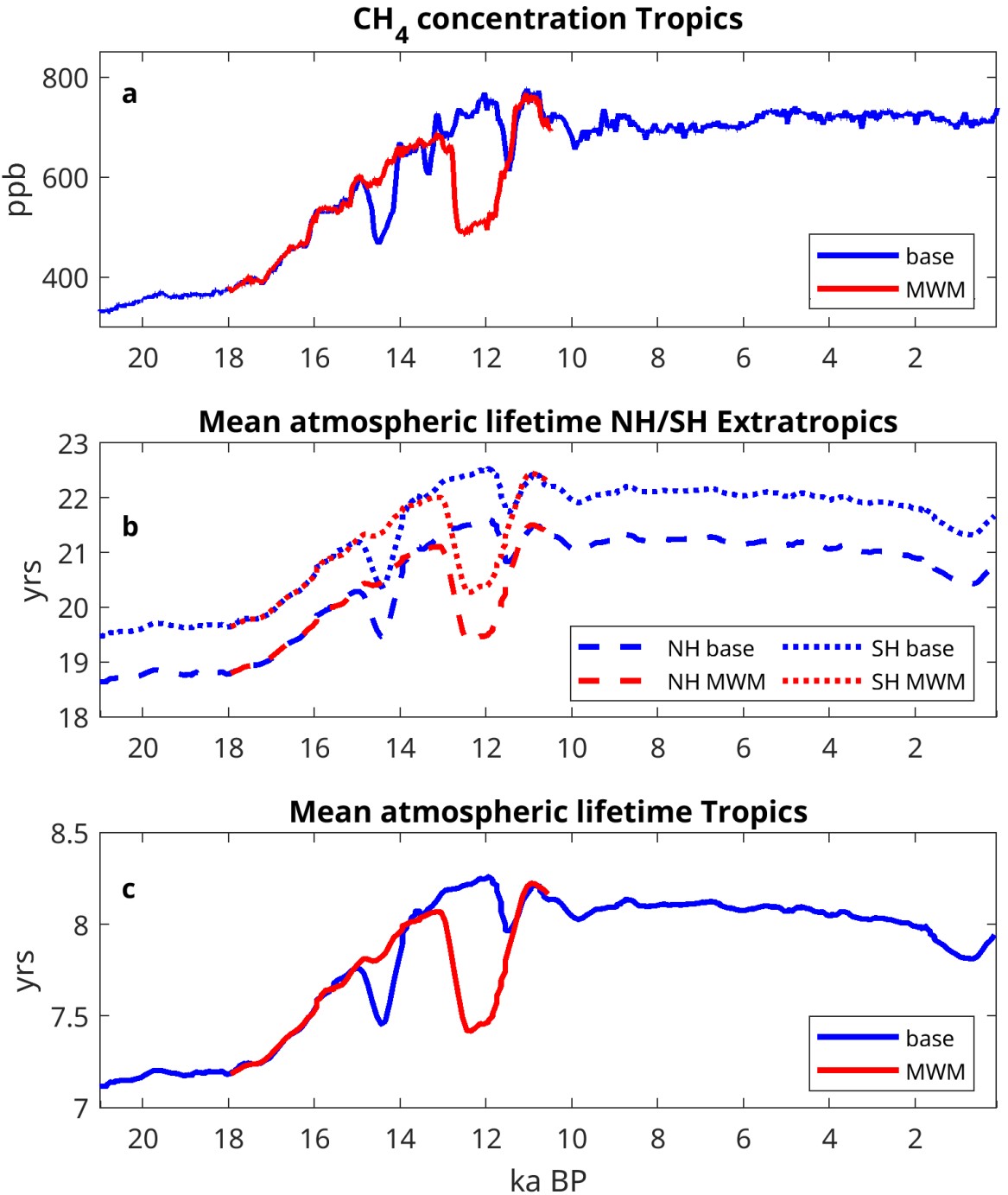

**Figure A1.** Tropical methane concentration (a) and mean atmospheric lifetime of methane in extratropical latitudes (b) and tropics (c).

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
