# Peer review of "Atmospheric methane since the LGM was driven by wetland sources"

_Climate of the Past, 2022_

## Referee Comment (RC1)

**Summary**

Kleinen et al. modelled the transient evolution of atmospheric methane mole fraction during the last deglaciation with a fully coupled Earth System model (ESM). As mentioned in the Introduction section of this paper, so far studies of glacial-interglacial methane cycles have been limited to simple box model exercises or ESM runs on steady state time slices. This manuscript provides a valiant first attempt to bridge the knowledge gap and provide valuable insights into the transient dynamics of Earth's methane system. Some of the highlight findings from this manuscript include

- Modeled prediction and improved mechanistic understanding about which part of the Earth's wetland region is most responsible for CH4 emissions associated with changes in AMOC during D-O#1 (OD-Bolling transition) and Younger Dryas-Preboreal transition. This provides a testable hypothesis for future CH4 interpolar gradient measurements from ice cores once the issue with *in situ* production in dusty Greenland ice (Lee et al., 2020) is dealt with. Furthermore, this in itself is also a good benchmark on how good this model is in predicting future CH4 emission (Kleinen et al., 2021).
- Constraints on how the CH4 lifetime (and oxidative capacity of the atmosphere) responded to CH4 emissions and how it can feedback back into the atmospheric CH4 burden during periods of abrupt CH4 rises
- Further emphasis on the importance of tropical wetlands for the global CH4 cycle, in agreement with top-down results from ice cores (e.g., Rhodes et al., 2015; Bock et al., 2017) and modern/recent top-down results (Lunt et al., 2019; Shaw et al., 2022).

I find the manuscript to be very well-written and enjoyable to read. The model input and results are discussed in sufficient details. I would highly recommend this manuscript for publication after some minor revisions. Kleinen et al. is sitting on a trove of important first results, and I think some of the additional data they already have from this experiment (such as interpolar gradient, simulated CH4 mole fraction in the tropics, latitude binned CH4 sink(s?), further details below) can be presented in a way that is more accessible and useful for future ice core/paleo CH4 studies. Furthermore, the discussion section of this manuscript is a bit short, and I think after conducting these experiments, Kleinen et al. is in a unique position to provide us with further insights about the glacial-interglacial methane dynamics and the role of some of the smaller CH4 sources (either quantitatively or qualitatively, will be elaborated further below).

**General comments**

One of the peculiar things about Kleinen et al. simulations is the relatively low fire emissions (Figure 3c, lower than 10 Tg CH4/yr during the Holocene), which I think disagree with most paleodata we have. From CH4 stable isotopes Bock et al. (2017) calculated certain acceptable solutions for total geological + fire CH4 emissions during the Holocene (Fig. 2 of their paper). If geological emissions is small (as constrained by the 14CH4 data and assumed in this study), then fire emissions has to be fairly large (on the order of 22-55 Tg CH4/yr) (Dyonisius et al., 2020) to balance and produce such a heavy d13C-CH4 and dD-CH4 signature recorded in ice core.

Measurements of other trace gases in ice cores that are co-emitted by fires (mainly CO, ethane and acetylene) (Wang et al., 2010; Nicewonger et al., 2020) also predict Holocene fire emissions (say around ~1000 CE) that is higher/comparable to modern day fire emissions (that is anthropogenic + wildfire total

emissions corresponding to ~40Tg CH4 per year). On the other hand, the global charcoal index (which is a bit more qualitative than trace gases in ice core) predict Holocene fire emissions that is a bit lower than modern (e.g., Marlon et al., 2008). So I think it is fair to say that the paleofire proxies are a bit all over the place, as they don't even agree with one another. However, even the charcoal record does not predict late Holocene fire emissions so low that it is less than ~1/3$^{rd}$ of total anthropogenic + wildfire emissions today.

Other than being a sizeable portion of the natural CH4 budget, fire emissions are obviously important because NOx, aerosol, CO, and NHMC (non methane hydrocarbon) emissions that affect the oxidative capacity of the atmosphere and CH4 lifetime. I understand that the low fire emissions used in Kleinen et al. simulations are simply the result from the well-cited SPITFIRE model (Lasslop et al., 2014) they used and there is nothing wrong with that. It might be prohibitedly expensive to rerun the transient experiment or conduct sensitivity analysis with larger fire emissions, I'm not sure. If a simple sensitivity analysis is not possible, I think at least Kleinen et al. should address this disagreement and maybe qualitatively discuss how their results would've changed if Holocene fire emissions as predicted by paleofire proxies mentioned above (and by extension maybe also LGM?) were a bit higher.

On a similar vein, the glacial-interglacial variability in CH4 uptake by soil seems a bit low (only +- couple of Tg CH4/yr over the whole deglaciation). Recent findings (for example Oh et al., 2020) suggest a much more dynamic soil uptake (at least in the high latitude) that can respond on decadal timescale to offset high arctic CH4 emissions associated with modern warming. Again, I do not expect Kleinen et al. to rerun the transient experiment with more sensitive/variable soil uptake parameter, but it would be nice if this is maybe qualitatively addressed in the discussion section.

In Section 2.3 where Kleinen et al. discuss atmospheric methane sink, it is also not immediately clear whether they explicitly include CH4 sink from reaction with chlorine (Allan et al., 2007). I presume that the chlorine sink is somewhere in there, considering ECHAM/MESSy model used in this study have been previously used to argue that the CH4 sink from tropospheric Cl reaction at the present is low (Gromov et al., 2018). It would be nice if this is explicitly clarified in the manuscript. Furthermore, although the Cl sink might be low, it has important effect on CH4 stable isotopes. If Kleinen et al. have a proper quantitative attribution to the temporal evolution of each individual CH4 sink (e.g., relative contributions from tropospheric Cl sink vs. reaction with OH, and other CH4 sinks such as stratosphere destruction, O(1)D)) during the deglaciation, an additional figure showing these parameters and short discussion would greatly benefit future studies of CH4 mole fraction and isotopes in ice core. It would also be highly beneficial to see a similar figure to figure 11a (CH4 flux by latitude band) but for CH4 sink/lifetime if such parameter exists and saved in the model runs. Finally, it is also okay if it these sink attributions are not explicitly available, but that should also be mentioned/discussed if Kleinen et al. think the relative importance of one vs. other can potentially change during the deglaciation.

Interpretations of CH4 studies from ice cores are often limited to 2 or 3 box models due to the practical limitation that we only have measurements from Greenland and Antarctic ice cores. A peculiar feature in some time slice paleo CH4 reconstruction from models (e.g., Murray et al., 2014) is that the CH4 mole fraction in the tropics is higher than in CH4 mole fraction in both poles during the LGM. Unfortunately, we cannot reliably measure and reconstruct tropical CH4 mole fraction from tropical/low-latitude alpine ice cores due to in situ production from organics in alpine ice cores. If CH4 mole fraction in the tropics (say in 30S to 30N lat bin) is indeed higher than the northern hemisphere, then obviously the 2, 3 box

model inversions commonly used in ice core studies (e.g., Chappellaz et al., 1997; Baumgartner et al., 2012) are inaccurate. The CH4 mole fraction in the tropics is a balance between CH4 emissions (which is highest in the tropics) and removal by OH (which is also highest in the tropics) – both of which can only be addressed with fully coupled CTM-ESM like the one used by Kleinen et al. It would greatly benefit the paleo-CH4 community, both experimentalists and modelers if Kleinen et al. can add to their figure 2 their reconstructed CH4 concentration over the tropics (despite the lack of data constraints) and provide some additional discussion about how reasonable they think their LGM simulation is (with focus on whether CH4 in the tropics is higher/lower than CH4 in Greenland during the LGM).

I'm also interested in the fact that in the transient simulations, the first abrupt CH4 spike seen by Kleinen et al. in both base and MWM scenario coincidentally occurred at 16 ka, concurrent with Heinrich stadial 1 (HS1) event. It might not be immediately obvious at first, but there is also a small and abrupt CH4 spike at 16 ka associated with HS1 (Rhodes et al., 2015). It has been argued that this small HS1 CH4 increase is due to southward movement in ITCZ activating/intensifying emissions from southern hemisphere wetlands (Seltzer et al., 2017; Rhodes et al., 2015).

In page 10 line 223, Kleinen et al. mentioned that they unfortunately do not have this equivalent HS1 event in their simulation, at least in term of AMOC signature. They argued that the CH4 rise they see at 16ka is actually D-O#1/OD-BO happening too early in the model. But I think the 16 ka coincidence warrants further investigation and discussion. I'm especially interested if Kleinen et al. think that there are any "Heinrich-like" events recorded somewhere in either the base or MWP simulations – for example, maybe the weakened AMOC state at ~15.5-14ka in the MWM simulation (figure 1)?. There are other indicators for Heinrich stadial on top of AMOC strength (like for example sea ice extent, Antarctic temperature, etc.) to check.

Heinrich events are particularly important in term of NH ice sheet evolution during the deglaciation. It would be very interesting to see additional discussion in this manuscript (doesn't have to be very long) on whether there is a Heinrich-like event in these simulations. If there is any, how this Heinrich like events affect the spatial distribution of CH4 emissions and if there is not, how the lack of 'Heinrich-like' event in the simulations affect the robustness of the interpretation (in term of say, sensitivity of CH4 emissions to AMOC changes driven by melting NH ice sheet).

Finally, if possible, I would highly recommend the authors to add the relatively simple time series data, especially the ones produced by their simulations (time series data to plot figure 3a, 3b, 3c, 4a,7b, 11a,11b) in the supplementary section of this paper, or somewhere online and easily accessible.

**Line comments**

Page 1 line 13: "four points in time". Not sure where the 4th abrupt CH4 transition is. I can only see 3 abrupt transitions during Termination 1, OD-BO CH4 rise, Allerod-YD CH4 drop, and YD-PB CH4 rise.

Page 2 line 41: "We investigated methane emissions [… ]5000 years *apart* from the LGM"; "apart" is not clear, I would change it to "*before* the LGM"

Page 3 line 69-70 "Methane emissions from wildfires …" word by word repeated in page 4 line 100. Alter one of either sentence by a little bit.

Page 5 line 140: I might have missed it, but I think "PFT" is not defined anywhere in this paper

Page 8 figure 1: The purpose of this figure is to provide broad overview of the model parameters and metric in term of T1 deglaciation. In my opinion, plotting some actual data on top would greatly help readers evaluate these model metrics. For example, for global mean temperature (fig. 1a) I think it would be nice to see Shakun et al. (2012) temperature reconstruction on top. It would be nice if atmospheric CO2 is plotted on the second y-axis of fig. 1c. Finally, another important metric that should be easily trackable in transient climate model simulation is mean ocean temperature (as integrator of various other metrics such as ice volume, sea level, AMOC strength etc). Mean ocean temperature can be plotted next to noble gas based mean ocean temperature reconstruction from ice cores (e.g., Baggenstos et al., 2019).

Page 9 fig2: As I mentioned above, CH4 concentration over the tropics would be very beneficial to plot here despite the lack of data constraints. Furthermore, the CH4 interpolar gradient (see Eq. 1 in Brook et al., 2000) is a commonly calculated analytical metric in ice core (Baumgartner et al., 2012; Sowers, 2010; Brook et al., 2000) and it would be great if it the CH4 interpolar gradient from the simulation runs can be calculated and presented in this figure. Finally, the missing Greenland ice core data at ~16 – 14ka is fair; but between 10-2ka, since it is the Holocene (which is not as dusty as the LGM), Greenland mole fraction from ice core is only minimally affected by in situ production (Lee et al., 2020). As such, I would highly recommend the authors to plot composite Greenland CH4 mole fraction by Beck et al. (2018).

Page 22 line 389: I would disagree with the assumption that all oceanic CH4 emission is geologic. There is a small amount of CH4 emissions from the open ocean due to decomposition/cycling of organic matter (Weber et al., 2019). This would likely have small impact on the overall result of the paper, but needs to be acknowledged.

**References**

Allan, W., Struthers, H., and Lowe, D. C.: Methane carbon isotope effects caused by atomic chlorine in the marine boundary layer: Global model results compared with Southern Hemisphere measurements, Journal of Geophysical Research-Atmospheres, 112, D04306, https://doi.org/10.1029/2006jd007369, 2007.

Baggenstos, D., Häberli, M., Schmitt, J., Shackleton, S. A., Birner, B., Severinghaus, J. P., Kellerhals, T., and Fischer, H.: Earth's radiative imbalance from the Last Glacial Maximum to the present, Proceedings of the National Academy of Sciences, 116, 14881–14886, 2019.

Baumgartner, M., Schilt, A., Eicher, O., Schmitt, J., Schwander, J., Spahni, R., Fischer, H., and Stocker, T. F.: High-resolution interpolar difference of atmospheric methane around the Last Glacial Maximum, 2012.

Beck, J., Bock, M., Schmitt, J., Seth, B., Blunier, T., and Fischer, H.: Bipolar carbon and hydrogen isotope constraints on the Holocene methane budget, Biogeosciences, 15, 7155–7175, https://doi.org/10.5194/bg-15-7155-2018, 2018.

Bock, M., Schmitt, J., Beck, J., Seth, B., Chappellaz, J., and Fischer, H.: Glacial/interglacial wetland, biomass burning, and geologic methane emissions constrained by dual stable isotopic CH4 ice core records, PNAS, 201613883, https://doi.org/10.1073/pnas.1613883114, 2017.

Brook, E. J., Harder, S., Severinghaus, J., Steig, E. J., and Sucher, C. M.: On the origin and timing of rapid changes in atmospheric methane during the Last Glacial Period, Global Biogeochem. Cycles, 14, 559–572, https://doi.org/10.1029/1999GB001182, 2000.

Chappellaz, J., Blunier, T., Kints, S., Dällenbach, A., Barnola, J.-M., Schwander, J., Raynaud, D., and Stauffer, B.: Changes in the atmospheric CH4 gradient between Greenland and Antarctica during the Holocene, J. Geophys. Res., 102, 15987–15997, https://doi.org/10.1029/97JD01017, 1997.

Dyonisius, M. N., Petrenko, V. V., Smith, A. M., Hua, Q., Yang, B., Schmitt, J., Beck, J., Seth, B., Bock, M., Hmiel, B., Vimont, I., Menking, J. A., Shackleton, S. A., Baggenstos, D., Bauska, T. K., Rhodes, R. H., Sperlich, P., Beaudette, R., Harth, C., Kalk, M., Brook, E. J., Fischer, H., Severinghaus, J. P., and Weiss, R. F.: Old carbon reservoirs were not important in the deglacial methane budget, Science, 367, 907–910, https://doi.org/10.1126/science.aax0504, 2020.

Gromov, S., Brenninkmeijer, C. A. M., and Jöckel, P.: A very limited role of tropospheric chlorine as a sink of the greenhouse gas methane, Atmospheric Chemistry and Physics, 18, 9831–9843, https://doi.org/10.5194/acp-18-9831-2018, 2018.

Kleinen, T., Gromov, S., Steil, B., and Brovkin, V.: Atmospheric methane underestimated in future climate projections, Environ. Res. Lett., 16, 094006, https://doi.org/10.1088/1748-9326/ac1814, 2021.

Lasslop, G., Thonicke, K., and Kloster, S.: SPITFIRE within the MPI Earth system model: Model development and evaluation, Journal of Advances in Modeling Earth Systems, 6, 740–755, https://doi.org/10.1002/2013MS000284, 2014.

Lee, J. E., Edwards, J. S., Schmitt, J., Fischer, H., Bock, M., and Brook, E. J.: Excess methane in Greenland ice cores associated with high dust concentrations, Geochimica et Cosmochimica Acta, 270, 409–430, 2020.

Lunt, M. F., Palmer, P. I., Feng, L., Taylor, C. M., Boesch, H., and Parker, R. J.: An increase in methane emissions from tropical Africa between 2010 and 2016 inferred from satellite data, Atmospheric Chemistry and Physics, 19, 14721–14740, https://doi.org/10.5194/acp-19-14721-2019, 2019.

Marlon, J. R., Bartlein, P. J., Carcaillet, C., Gavin, D. G., Harrison, S. P., Higuera, P. E., Joos, F., Power, M. J., and Prentice, I. C.: Climate and human influences on global biomass burning over the past two millennia, Nature Geoscience, 1, 697–702, https://doi.org/10.1038/ngeo313, 2008.

Murray, L. T., Mickley, L. J., Kaplan, J. O., Sofen, E. D., Pfeiffer, M., and Alexander, B.: Factors controlling variability in the oxidative capacity of the troposphere since the Last Glacial Maximum, Atmos. Chem. Phys, 14, 3589–3622, 2014.

Nicewonger, M. R., Aydin, M., Prather, M. J., and Saltzman, E. S.: Extracting a History of Global Fire Emissions for the Past Millennium From Ice Core Records of Acetylene, Ethane, and Methane, Journal of Geophysical Research: Atmospheres, 125, e2020JD032932, https://doi.org/10.1029/2020JD032932, 2020.

Oh, Y., Zhuang, Q., Liu, L., Welp, L. R., Lau, M. C. Y., Onstott, T. C., Medvigy, D., Bruhwiler, L., Dlugokencky, E. J., Hugelius, G., D'Imperio, L., and Elberling, B.: Reduced net methane emissions due to

microbial methane oxidation in a warmer Arctic, Nature Climate Change, 10, 317–321, https://doi.org/10.1038/s41558-020-0734-z, 2020.

Rhodes, R. H., Brook, E. J., Chiang, J. C., Blunier, T., Maselli, O. J., McConnell, J. R., Romanini, D., and Severinghaus, J. P.: Enhanced tropical methane production in response to iceberg discharge in the North Atlantic, Science, 348, 1016–1019, 2015.

Seltzer, A. M., Buizert, C., Baggenstos, D., Brook, E. J., Ahn, J., Ji-Woong, Y., and Severinghaus, J. P.: Does $\delta$ 18 O of O 2 record meridional shifts in tropical rainfall?, Climate of the Past, 13, 1323, 2017.

Shakun, J. D., Clark, P. U., He, F., Marcott, S. A., Mix, A. C., Liu, Z., Otto-Bliesner, B., Schmittner, A., and Bard, E.: Global warming preceded by increasing carbon dioxide concentrations during the last deglaciation, Nature, 484, 49–54, https://doi.org/10.1038/nature10915, 2012.

Shaw, J. T., Allen, G., Barker, P., Pitt, J. R., Pasternak, D., Bauguitte, S. J.-B., Lee, J., Bower, K. N., Daly, M. C., Lunt, M. F., Ganesan, A. L., Vaughan, A. R., Chibesakunda, F., Lambakasa, M., Fisher, R. E., France, J. L., Lowry, D., Palmer, P. I., Metzger, S., Parker, R. J., Gedney, N., Bateson, P., Cain, M., Lorente, A., Borsdorff, T., and Nisbet, E. G.: Large Methane Emission Fluxes Observed From Tropical Wetlands in Zambia, Global Biogeochemical Cycles, 36, e2021GB007261, https://doi.org/10.1029/2021GB007261, 2022.

Sowers, T.: Atmospheric methane isotope records covering the Holocene period, Quaternary Science Reviews, 29, 213–221, https://doi.org/10.1016/j.quascirev.2009.05.023, 2010.

Wang, Z., Chappellaz, J., Park, K., and Mak, J. E.: Large Variations in Southern Hemisphere Biomass Burning During the Last 650 Years, Science, 330, 1663–1666, https://doi.org/10.1126/science.1197257, 2010.

Weber, T., Wiseman, N. A., and Kock, A.: Global ocean methane emissions dominated by shallow coastal waters, Nat Commun, 10, 4584, https://doi.org/10.1038/s41467-019-12541-7, 2019.

---

## Author Response (AR1)

**Reply to the reviews**

by Thomas Kleinen

We thank both reviewers for their very constructive comments. As you will see in the following, we have taken up most of their suggestions when drafting this revised version of our manuscript, though deviating in the interest of brevity in a few places.

The main change to the revised manuscript is that we extended the discussion section, discussing both the overall methane budget and a number of fluxes in more detail. We also modified a number of the Figures showing time-series, following suggestions by the reviewers.

In the following, we discuss point-by-point our response to the reviewers' comments. The reviewer's comment will be in blue, our previous reply to the reviewers in red, and the discussion of revisions undertaken in black. All line numbers refer to the also submitted document showing the changes between our original submission and the revised version.

**Reply to Anonymous Referee #1**

by Thomas Kleinen

We very much thank the reviewer for taking the time to review our manuscript. We especially appreciate the very valuable suggestions for improving the manuscript which appear to come from a perspective that is quite distinct from our modelling focus. I have included the reviewer's comments in blue colour, while our reply is in red.

**Summary**

Kleinen et al. modelled the transient evolution of atmospheric methane mole fraction during the last deglaciation with a fully coupled Earth System model (ESM). As mentioned in the Introduction section of this paper, so far studies of glacial-interglacial methane cycles have been limited to simple box model exercises or ESM runs on steady state time slices. This manuscript provides a valiant first attempt to bridge the knowledge gap and provide valuable insights into the transient dynamics of Earth's methane system. Some of the highlight findings from this manuscript include

- Modeled prediction and improved mechanistic understanding about which part of the Earth's wetland region is most responsible for $CH_4$ emissions associated with changes in AMOC during D-O#1 (OD-Bolling transition) and Younger Dryas-Preboreal transition. This provides a testable hypothesis for future $CH_4$ interpolar gradient measurements from ice cores once the issue with in situ production in dusty Greenland ice (Lee et al., 2020) is dealt with. Furthermore, this in itself is also a good benchmark on how good this model is in predicting future $CH_4$ emission (Kleinen et al., 2021).

- Constraints on how the $CH_4$ lifetime (and oxidative capacity of the atmosphere) responded to $CH_4$ emissions and how it can feedback back into the atmospheric $CH_4$ burden during periods of abrupt $CH_4$ rises

- Further emphasis on the importance of tropical wetlands for the global $CH_4$ cycle, in agreement with top-down results from ice cores (e.g., Rhodes et al., 2015; Bock et al., 2017) and modern/recent top-down results (Lunt et al., 2019; Shaw et al., 2022).

I find the manuscript to be very well-written and enjoyable to read. The model input and results are discussed in sufficient details. I would highly recommend this manuscript for publication after some minor revisions. Kleinen et al. is sitting on a trove of important first results, and I think some of the additional data they already have from this experiment (such as interpolar gradient, simulated $CH_4$ mole fraction in the tropics, latitude binned $CH_4$ sink(s)?, further details below) can be presented in a way that is more accessible and useful for future ice core/paleo $CH_4$ studies. Furthermore, the discussion section of this manuscript is a bit short, and I think after conducting these experiments,

Kleinen et al. is in a unique position to provide us with further insights about the glacial-interglacial methane dynamics and the role of some of the smaller CH4 sources (either quantitatively or qualitatively, will be elaborated further below).

Thank you very much. We appreciate the fact that the reviewer seems to regard our results highly and will be happy to take his or her suggestions to improve our manuscript.

**General comments**
One of the peculiar things about Kleinen et al. simulations is the relatively low fire emissions (Figure 3c, lower than 10 Tg CH4/yr during the Holocene), which I think disagree with most paleodata we have. From CH4 stable isotopes Bock et al. (2017) calculated certain acceptable solutions for total geological + fire CH4 emissions during the Holocene (Fig. 2 of their paper). If geological emissions is small (as constrained by the 14CH4 data and assumed in this study), then fire emissions has to be fairly large (on the order of 22-55 Tg CH4/yr) (Dyonisius et al., 2020) to balance and produce such a heavy d13C-CH4 and dD-CH4 signature recorded in ice core. Measurements of other trace gases in ice cores that are co-emitted by fires (mainly CO, ethane and acetylene) (Wang et al., 2010; Nicewonger et al., 2020) also predict Holocene fire emissions (say around ~1000 CE) that is higher/comparable to modern day fire emissions (that is anthropogenic + wildfire total emissions corresponding to ~40Tg CH4 per year). On the other hand, the global charcoal index (which is a bit more qualitative than trace gases in ice core) predict Holocene fire emissions that is a bit lower than modern (e.g., Marlon et al., 2008). So I think it is fair to say that the paleofire proxies are a bit all over the place, as they don't even agree with one another. However, even the charcoal record does not predict late Holocene fire emissions so low that it is less than ~1/3 rd of total anthropogenic + wildfire emissions today.
Other than being a sizeable portion of the natural CH4 budget, fire emissions are obviously important because NOx, aerosol, CO, and NHMC (non methane hydrocarbon) emissions that affect the oxidative capacity of the atmosphere and CH4 lifetime. I understand that the low fire emissions used in Kleinen et al. simulations are simply the result from the well-cited SPITFIRE model (Lasslop et al., 2014) they used and there is nothing wrong with that. It might be prohibitedly expensive to rerun the transient experiment or conduct sensitivity analysis with larger fire emissions, I'm not sure. If a simple sensitivity analysis is not possible, I think at least Kleinen et al. should address this disagreement and maybe qualitatively discuss how their results would've changed if Holocene fire emissions as predicted by paleofire proxies mentioned above (and by extension maybe also LGM?) were a bit higher. On a similar vein, the glacial-interglacial variability in CH4 uptake by soil seems a bit low (only +- couple of Tg CH4/yr over the whole deglaciation). Recent findings (for example Oh et al., 2020) suggest a much more dynamic soil uptake (at least in the high latitude) that can respond on decadal timescale to offset high arctic CH4 emissions associated with modern warming. Again, I do not expect Kleinen et al. to rerun the transient experiment with more sensitive/variable soil uptake parameter, but it would be nice if this is maybe qualitatively addressed in the discussion section.

The reviewer touches upon two important points we apparently did not address sufficiently in our manuscript, the emissions from wildfires and the soil uptake.
The relatively low emissions from wildfires are, as the reviewer assumes, the result of applying the SPITFIRE model. The SPITFIRE model is – by ESM standards – a relatively sophisticated and well-published fire model. It may, however, not be ideal for this particular application. In the development of the model, the main focus lay on the description of fires in the modern period, which are mainly ignited by humans. The description of wildfires with ignition by natural factors, mainly lightning, is somewhat less sophisticated, however, and this may well be the reason for the relatively low fire emissions in glacial climate. In particular, the model assumes that lightning does not change in glacial climate – an assumption that is likely untrue. However we do not have a better description of lightning strike available, and the lightning model we use for the estimation of

lightning NOx does not show more lightning strikes in glacial than in preindustrial climate. The reviewer is quite right, unfortunately it is prohibitively expensive to re-run the model experiment (it takes 3-4 months and several 10k€). Thus we will have to discuss this issue more carefully than we have done in the reviewed manuscript, which we will do in the revised version.

With regard to the uptake of methane by soils, we are aware of the Oh et al. results. In our model, however, the soil uptake of methane is not limited by the rate at which microbes are able to process the methane, but it is rather limited by the rate of diffusion of methane from the atmosphere into the surface soil. Thus we are not able to model this directly, but we will certainly discuss this in the revised version of our manuscript.

In the revised manuscript, we extended the discussion section. A section discussing fire fluxes was added at lines 463 – 477. Here we discuss influences on fire occurrence, as well as comparison against proxies. Furthermore, we discuss factors determining soil methane uptake and the potential influence of the Oh et al flux observation on lines 494 – 506.
Finally, we added a longer section near the beginning of the discussion section (lines 402 – 419) where we discuss the constraints of the deglacial methane budget, and which magnitudes of fluxes appear feasible in the light of reconstructed atmospheric $CH_4$ and methane chemistry, taking up the Bock et al. (2017) estimate of combined geological and wildfire emissions, as well as modern estimates (Saunois et al. 2020) of present-day fluxes..

In Section 2.3 where Kleinen et al. discuss atmospheric methane sink, it is also not immediately clear whether they explicitly include CH4 sink from reaction with chlorine (Allan et al., 2007). I presume that the chlorine sink is somewhere in there, considering ECHAM/MESSy model used in this study have been previously used to argue that the CH4 sink from tropospheric Cl reaction at the present is low (Gromov et al., 2018). It would be nice if this is explicitly clarified in the manuscript. Furthermore, although the Cl sink might be low, it has important effect on CH4 stable isotopes. If Kleinen et al. have a proper quantitative attribution to the temporal evolution of each individual CH4 sink (e.g., relative contributions from tropospheric Cl sink vs. reaction with OH, and other CH4 sinks such as stratosphere destruction, O(1)D)) during the deglaciation, an additional figure showing these parameters and short discussion would greatly benefit future studies of CH4 mole fraction and isotopes in ice core. It would also be highly beneficial to see a similar figure to figure 11a (CH4 flux by latitude band) but for CH4 sink/lifetime if such parameter exists and saved in the model runs. Finally, it is also okay if it these sink attributions are not explicitly available, but that should also be mentioned/discussed if Kleinen et al. think the relative importance of one vs. other can potentially change during the deglaciation.

The methane sink formulation we are using is a highly parameterised simplification derived from the ECHAM/Messy model. As a result, we can only determine the combined effect of all the different sink terms, but not the individual contributions. We will extend the discussion of the methane sink to the different terms, though. With regard to the Figure of methane sink by latitude, we will try to include such a Figure in the revised manuscript. I believe all the necessary output should be available.

In the revised manuscript, we have added an additional Figure in the Appendix, showing methane lifetimes for latitudes 90°S-30°S, 30°S-30°N and 30°N-90°N. We have also extend the discussion of the methane sink, especially in terms of different sink mechanisms, on lines 506 – 517.

Interpretations of CH4 studies from ice cores are often limited to 2 or 3 box models due to the practical limitation that we only have measurements from Greenland and Antarctic ice cores. A peculiar feature in some time slice paleo CH4 reconstruction from models (e.g., Murray et al., 2014) is that the CH4 mole fraction in the tropics is higher than in CH4 mole fraction in both poles during

the LGM. Unfortunately, we cannot reliably measure and reconstruct tropical CH4 mole fraction from tropical/low-latitude alpine ice cores due to in situ production from organics in alpine ice cores. If CH4 mole fraction in the tropics (say in 30S to 30N lat bin) is indeed higher than the northern hemisphere, then obviously the 2, 3 box model inversions commonly used in ice core studies (e.g., Chappellaz et al., 1997; Baumgartner et al., 2012) are inaccurate. The CH4 mole fraction in the tropics is a balance between CH4 emissions (which is highest in the tropics) and removal by OH (which is also highest in the tropics) – both of which can only be addressed with fully coupled CTM-ESM like the one used by Kleinen et al. It would greatly benefit the paleo-CH4 community, both experimentalists and modelers if Kleinen et al. can add to their figure 2 their reconstructed CH4 concentration over the tropics (despite the lack of data constraints) and provide some additional discussion about how reasonable they think their LGM simulation is (with focus on whether CH4 in the tropics is higher/lower than CH4 in Greenland during the LGM).

Thank you very much for this excellent suggestion – we didn't think of it as it is never discussed in the literature, but it's easily feasible and may be quite valuable.

We have added an additional Figure in the Appendix, showing the mean tropical $CH_4$ concentration, also discussing it on lines 259 – 260.

I'm also interested in the fact that in the transient simulations, the first abrupt CH4 spike seen by Kleinen et al. in both base and MWM scenario coincidentally occurred at 16 ka, concurrent with Heinrich stadial 1 (HS1) event. It might not be immediately obvious at first, but there is also a small and abrupt CH4 spike at 16 ka associated with HS1 (Rhodes et al., 2015). It has been argued that this small HS1 CH4 increase is due to southward movement in ITCZ activating/intensifying emissions from southern hemisphere wetlands (Seltzer et al., 2017; Rhodes et al., 2015).
In page 10 line 223, Kleinen et al. mentioned that they unfortunately do not have this equivalent HS1 event in their simulation, at least in term of AMOC signature. They argued that the CH4 rise they see at 16ka is actually D-O#1/OD-BO happening too early in the model. But I think the 16 ka coincidence warrants further investigation and discussion. I'm especially interested if Kleinen et al. think that there are any "Heinrich-like" events recorded somewhere in either the base or MWP simulations – for example, maybe the weakened AMOC state at ~15.5-14ka in the MWM simulation (figure 1)?. There are other indicators for Heinrich stadial on top of AMOC strength (like for example sea ice extent, Antarctic temperature, etc.) to check.
Heinrich events are particularly important in term of NH ice sheet evolution during the deglaciation. It would be very interesting to see additional discussion in this manuscript (doesn't have to be very long) on whether there is a Heinrich-like event in these simulations. If there is any, how this Heinrich like events affect the spatial distribution of CH4 emissions and if there is not, how the lack of 'Heinrich-like' event in the simulations affect the robustness of the interpretation (in term of say, sensitivity of CH4 emissions to AMOC changes driven by melting NH ice sheet).

Strictly speaking our current experimental setup is not capable of producing Heinrich events, as a Heinrich event in our current understanding is a dynamical interplay of ocean and ice-sheet dynamics. With prescribed ice sheets, it is not possible to obtain this, a dynamically coupled ice sheet model is required instead (see Ziemen et al. (2019, Clim. Past., https://doi.org/10.5194/cp-15-153-2019) for an example). We are currently working on this, but we are not there yet.
Having said that, we do obtain events resembling Heinrich events in our current experiment, the Younger Dryas transition in our MWM experiment being an example of one, and there may be other events showing some of the characteristics of H events in our deglaciation experiment – we will look at more diagnostics before submitting the revised manuscript, and will certainly extend the discussion of H events for the revision.
We discuss H events in the discussion section, lines 388 – 401. No, we do not see evidence of further H events in our model, with the exception of extremely short excursions. The BA to YD

transition we demonstrate in experiment MWM, however, shows a pattern that we would expect to be very similar to any other H event, though obviously dependent on the background state.

Finally, if possible, I would highly recommend the authors to add the relatively simple time series data, especially the ones produced by their simulations (time series data to plot figure 3a, 3b, 3c, 4c, 7a,7b, 11a,11b) in the supplementary section of this paper, or somewhere online and easily accessible.

Thank you. Yes, it is indeed planned to make the full output of the model available on the Earth System Grid. Processing of the output is currently ongoing but rather time-consuming as it involves very large amounts of data, so publication of the data set may happen at a slightly later date than publication of the paper. We had not yet considered also publishing the aggregated time-series output, but we will certainly rethink that as it makes the output accessible to a wider audience.
Full model output will be publically available from the Earth System Grid, a further set of time-series data has been published on Zenodo. We hope that everything interesting is available now, but are open to suggestions.

**Line comments**
Page 1 line 13: "four points in time". Not sure where the 4th abrupt CH4 transition is. I can only see 3 abrupt transitions during Termination 1, OD-BO CH4 rise, Allerod-YD CH4 drop, and YD-PB CH4 rise.
Oooops, thanks for pointing that out. Will correct.
Done. Line 14.

Page 2 line 41: "We investigated methane emissions [... ]5000 years apart from the LGM"; "apart" is not clear, I would change it to "before the LGM"
Thanks. We will correct that.
Done. Line 43.

Page 3 line 69-70 "Methane emissions from wildfires ..." word by word repeated in page 4 line 100. Alter one of either sentence by a little bit.
Thanks. We will correct that.
Done. Line 72 - 73.

Page 5 line 140: I might have missed it, but I think "PFT" is not defined anywhere in this paper
That is entirely possible. We will check and correct.
Done. Line 144.

Page 8 figure 1: The purpose of this figure is to provide broad overview of the model parameters and metric in term of T1 deglaciation. In my opinion, plotting some actual data on top would greatly help readers evaluate these model metrics. For example, for global mean temperature (fig. 1a) I think it would be nice to see Shakun et al. (2012) temperature reconstruction on top. It would be nice if atmospheric CO2 is plotted on the second y-axis of fig. 1c. Finally, another important metric that should be easily trackable in transient climate model simulation is mean ocean temperature (as integrator of various other metrics such as ice volume, sea level, AMOC strength etc). Mean ocean temperature can be plotted next to noble gas based mean ocean temperature reconstruction from ice cores (e.g., Baggenstos et al., 2019).
Thanks for the suggestion. We will incorporate this.
Mostly Done. We included temperatures from Shakun et al. (2014), as well as Osman et al. (2021) in Figure 1a, also discussed on lines 202 - 205. $CO_2$ is also plotted in Fig. 1c. However, we were unable to also show mean ocean temperature, as we are very much unsure on how to compare it to reference data and think of it as less relevant to the majority of the audience.

Page 9 fig2: As I mentioned above, CH4 concentration over the tropics would be very beneficial to plot here despite the lack of data constraints. Furthermore, the CH4 interpolar gradient (see Eq. 1 in Brook et al., 2000) is a commonly calculated analytical metric in ice core (Baumgartner et al., 2012; Sowers, 2010; Brook et al., 2000) and it would be great if it the CH4 interpolar gradient from the simulation runs can be calculated and presented in this figure. Finally, the missing Greenland ice core data at ~16 – 14ka is fair; but between 10-2ka, since it is the Holocene (which is not as dusty as the LGM), Greenland mole fraction from ice core is only minimally affected by in situ production (Lee et al., 2020). As such, I would highly recommend the authors to plot composite Greenland CH4 mole fraction by Beck et al. (2018).
Thanks for the suggestion. We will extend the Figure.
Done. We have added a third panel to Figure 2, showing the methane gradient between Greenland and Antarctica. Due to lack of space here, we moved the Figure showing tropical $CH_4$ to the Appendix. This is discussed on Lines 256 – 259. We also changed the Greenland reference data to Beck (2018) to extend the timeseries available for comparison. Figure 2b and lines 251 – 252.

Page 22 line 389: I would disagree with the assumption that all oceanic CH4 emission is geologic. There is a small amount of CH4 emissions from the open ocean due to decomposition/cycling of organic matter (Weber et al., 2019). This would likely have small impact on the overall result of the paper, but needs to be acknowledged.
Thanks for that correction, our formulation was a little hasty. We will correct this.
Corrected in lines 442 - 443.

**Reply to Anonymous Referee #2**

by Thomas Kleinen

We very much thank the reviewer for taking the time to review our manuscript. I have included the reviewer's comments in blue colour, while our reply is in black.

**Summary:**
This paper by Kleinen et al. makes use of a CMIP6-generation Earth System Model, MPI- ESM, with an interactive methane cycle to investigate changes in the methane budget between the last glacial maximum and the pre-industrial period. The model includes interactive emission schemes for many of the natural emission sources relevant for the time period of interest and includes a parametrised approach for the atmospheric methane sink. Using novel, and for the first time, transient simulations, they focus particularly on the rapid changes in the methane cycle occurring during deglaciation.
The paper is well organised, with clear and sufficient detail for the reader to understand the model set up and the results. It is well written and made for an enjoyable and interesting read. More importantly, this study represents a significant step change in model capability, model setup, and an advancement in the state-of-the-art, particularly in relation to running transient simulations from the last glacial maximum to the present day. To date, other studies addressing changes in the methane cycle over this time period have either used simple models or timeslice simulations.
We very much thank the reviewer for her or his praise of our manuscript.

Below, I have some minor general and/or specific comments. However, I would unreservedly recommend that the manuscript be published.

**General comments:**
You say that the tropical wetland extent is overestimated in the model. Can you comment on how much that overestimate influences your conclusions regarding the role of tropical wetlands in driving the changes?
In Kleinen et al (2020), we compare the modelled inundation to remote sensing data by Prigent et al. (2012), with areas by Prigent about 30% smaller than the model estimate. However, this remote sensing estimate very likely is an underestimate, as it relies on optical sensing, combined with radar estimates in a band that cannot penetrate the tree canopy. As a significant part of the tropical wetland area is located in rainforests, the likelihood that the true extent is larger than in this data set is rather high. Thus the model estimate may be an overestimate, but as the uncertainty in the remote-sensing data is higher than we would like, it the model might also be closer to the true extent than this comparison would suggest.
Having said that, the modelled latitudinal distribution of present-day wetland emissions, as well as the total emissions, is similar to data-based estimates. We thus assume that we are "close enough" to the true extent, but we cannot prove it.
The emissions from tropical wetlands are the largest single factor in the methane balance, making up at least 50% of emissions at all times. Also, the absolute increase in emissions is largest for tropical wetlands. However, the increase in emissions from NH extratropical wetlands is not much smaller than the increase in tropical emissions in absolute terms, implying that the ranking would change if our estimated wetland extent proves to be significantly too large.

We will discuss this in the revised version of our manuscript and word it more carefully.
We start the discussion section in the revised manuscript with an extended paragraph on modelling assumptions, as well as a few paragraphs discussing how the modelled fluxes evaluate against other sources. Here we also discuss the wetland fluxes, lines 453 – 462.

As a scientist with an interest in the more contemporary period and future projections, I'd be keen for the manuscript to include some discussion on the implications of this study for future projections of methane and the role of tropical wetland sources. You say that for the purpose of accounting for the soil uptake, you prescribe the atmospheric concentration of methane. Can you comment on the potential impact on model performance that would arise if soil uptake was coupled to the modelled concentration?

In Kleinen et al. (2021) we have already published an assessment of future methane concentrations under a number of the SSP scenarios. We would very much prefer not to duplicate our previous publication and therefore refrained from including any results for beyond preindustrial.

With regard to the soil uptake of methane, the difference between using prescribed or modelled methane concentrations is relatively small, as long as the difference between modelled and reconstructed concentration is small. The impact would thus be most significant at mid-Holocene when the difference between reconstructed and modelled concentration is largest. The modelled methane uptake at that time would have been slightly higher, thus decreasing atmospheric methane, if we had used the model methane concentration.

We will discuss this in the revised manuscript.

Since we have written an entire paper on future methane emissions and concentrations, we have not extended the discussion in that direction, we had the impression it would not have improved the current manuscript. We did, however, extend the discussion of the soil sink of methane, also addressing the difference in soil uptake due to the use of prescribed $CH_4$ there. Lines 494 – 505.

**Specific comments:**

Page 3, line 89: Change "is produces" to "is produced"

Thanks for pointing this out, will be corrected.

Done. Line 93.

Line 161: For use of "CI" in the first instance, please write out in full with abbreviation. Thereafter, CI is okay to use.

Thanks, will be included.

Corrected. Line 167.

Line 195: On first use, please write out "AMOC" in full with abbreviation

We will correct this oversight.

Oversight corrected. Line 190.

Lines 196 and 199: As a reviewer whose main expertise is more in the contemporary period, it would be useful to explain what is meant by "1a" and "1b" when referring to the meltwater pulse. If they are simply referring to the different transitions in the AMOC which occur, perhaps these could either be labelled in the figure (and with addition to figure caption) or made more explicit in the text.

Thank you very much for pointing this out, we neglected to explain this properly in our manuscript. The meltwater pulses 1a and 1b were (to our knowledge) first identified by Fairbanks in reconstructions of deglaciual sea level rise from Barbados corals (R.G. Fairbanks (1989), A 17,000-year glacio-eustatic sea level record: influence of glacial melting rates on the Younger Dryas event and deep-ocean circulation, Nature, 342, 637-642, doi:10.1038/342637a0) and occurred at 12000 and 9500 radiocarbon years BP.

As the sea level history is not actually part of the model forcing, we cannot easily label these in the Figure. We will, however, discuss it more carefully in the text, referring the reader to Fairbanks.

We discuss the meltwater pulses in the text, making clearer to the audience what they refer to. Line 207.

Caption for Table 1: Suggest that you change "timeslices" to "time periods"
Thanks. Will be changed in revised manuscript.
Thanks. Done. See Table legend.

---

## Author Response (AR2)

**Reply to the review**

by Thomas Kleinen

We thank the reviewer – again – for her or his helpful comments on the revised version of our manuscript. For the (hopefully) final revision of our manuscript, we have taken up most of the suggestions.
In the following, we discuss point-by-point our response to the reviewer's comments. The reviewer's comment will be in blue, and our response will be in black. All line numbers refer to the track-changes document.

**Reply to Anonymous Referee #2**

Kleinen et al. have done a great job addressing my previous comments in a satisfactory manner. They also done a great job in elaborating and adding several important discussion points to their paper. This study provides a novel first attempt to model the transient evolution of methane during the last deglaciation. From modeling approach, it brought forward many important results and ideas, mainly the importance of tropical wetlands in driving the bulk of deglacial methane rise, the role of methane emissions from shelf areas, the evolution of atmospheric methane lifetime during the deglacial transition. As such I would highly recommend this manuscript for publications.

We thank the reviewer for their appreciation of our manuscript.

Here are some minor notes I have that might be useful in polishing up the paper:

Page 1 line 13: "Between the last glacial [...] doubling in concentration during those 11000 yrs." I would cite Figure 2 here just to orient readers who are not extremely familiar with deglacial methane evolution.
Added reference to Fig. 2 on line 13.

Page 2 line 27: "Or they used a more detailed [...]." Here 'they' refer to non-box model (so non transient) time slice studies but it is not immediately clear. To clarify and draw the contrast against the box model studies I would just say something like "An alternative approach to box models would be ... "
We reformulated the text along the lines the reviewer suggested, though using a slightly different wording, Lines 25 – 27: Many of these studies were performed with strongly simplified... The alternative to very simplified models were studies using models with more detailed...

Page 3 line 60-65: This opening paragraph is quite redundant as every aspect is elaborated further in details below. I would consider removing it entirely.
We thank the reviewer for the suggestion but take the liberty to deviate from this suggestion: This paragraph serves to set the stage for the following more detailed description, we believe it makes the text more accessible. We therefore opted to keep it.

Page 4 line 103: "assuming no human population before 12 ka BP". I'm sure with regards to methane emissions, the effect of no human population vs. small human population at the time is fairly negligible, but this statement jumps out as obviously untrue as humans were clearly around in the LGM. Maybe just explicitly say something like "we assume no human population because the effect of ~2 million stone age people on fire emissions is small" to avoid confusion. I'm not an expert in human population during the LGM, just happen to stumble across the 2 million number from Gautney and Holliday (2015).
Thank you, that is obviously correct. We reformulated (line 104): ... (assuming no human impact on

fires before 12 ka BP due to small population size and using Klein Goldewijk et al. (2017) afterwards)…

Page 6 line 178-181: This part I think fit better with the previous section, maybe just add it to section 2.1 and add "geological emissions" to the title of section 2.1
We did exactly that, moving the paragraph to lines 109-113. This passage now reads: Finally, methane emissions from geological sources are prescribed using a spatial distribution from Etiope (2015), but scaled down to give total geological methane emissions of 5 T gCH 4 yr−1 , as Petrenko et al. (2017) and Hmiel et al. (2020) show from ice-core data that geological emissions larger than this value are not possible for either the Younger Dryas or the preindustrial period.

Page 11 line 247: "when dust accumulation". I think the word 'concentration' fits better here than 'accumulation.'
Thanks, corrected.

Page 11 line 255: "Finally, the tropical methane concentration [...] throughout the experiment." Might be worth mentioning that this result is in contrast to some earlier time slice studies like for example very clearly shown in figure 2b of Valdes et al. (2005) where methane concentration during the LGM is highest in the tropics.
Lines 260 - 262 now read: Finally, the tropical methane concentration (Fig. A1a) stays in between the values for Antarctica and Greenland throughout the experiment, in contrast to previous studies showing LGM concentrations highest in the tropics (Valdes et al., 2005).

Page 11 line 257-274. This is arguably the crux of the paper. Methane is driven by wetlands, which is driven by AMOC. Here the authors have a very nice description of the model results where methane emissions from wetlands respond to AMOC. What is a bit lacking (I understand this is later elaborated in section 3.3) is maybe a brief explanation of why (process- wise) wetland emissions are coupled so strongly to AMOC.
We have added a brief sentence here (lines 272 – 274):  Here, the AMOC collapse leads to a significant decrease in NH temperatures around the Atlantic ocean, in turn leading to decreased evaporation and thus decreased precipitation, thereby decreasing wetland areas and methane production in the NH tropics.

Page 19 line 355 to page 21 line 370. This section again pertains mostly to AMOC, Bolling-Allerod and YD transitions. I personally think this should be on Section 3.3 rather than a section about Holocene.
Here, we once again beg to differ, as we believe that this text should really be in this location. However, the reviewer drew our attention to the fact that the title of this section may have been less than ideal – we therefore renamed the section to "Regional distribution of methane fluxes over time" (Section 3.5, line 352), as it is really less about the Holocene and more about the regional flux distribution.

As you can see, we addressed most of the reviewer's comments, hopefully in a satisfactory fashion. We hope that the manuscript can now be accepted for publication.